# Effects of drought on the abundance and distribution of non-breeding shorebirds in central California, USA

Blake A. Barbaree[1]*, Matthew E. Reiter[1], Catherine M. Hickey[1], Khara M. Strum[2], Jennifer E. Isola[3], Scott Jennings[4], L. Max Tarjan[5], Cheryl M. Strong[6], Lynne E. Stenzel[1], W. David Shuford[1]

**1** Point Blue Conservation Science, Petaluma, California, United States of America, **2** Audubon California, Sacramento, California, United States of America, **3** United States Fish and Wildlife Service, Sacramento National Wildlife Refuge Complex, Willows, California, United States of America, **4** Audubon Canyon Ranch, Marshall, California, United States of America, **5** San Francisco Bay Bird Observatory, Milpitas, California, United States of America, **6** United States Fish and Wildlife Service, Newport Field Office, Newport, Oregon, United States of America

* bbarbaree@pointblue.org

**Data Availability Statement:** All relevant data are within the manuscript and its Supporting Information files.

## Abstract

Conservation of migratory species requires anticipating the potential impacts of extreme climatic events, such as extreme drought. During drought, reduced habitat availability for shorebirds creates the potential for changes in their abundance and distribution, in part because many species are highly mobile and rely on networks of interior and coastal habitats. Understanding how shorebirds responded to a recent drought cycle that peaked from 2013 to 2015 in central California, USA, will help optimize management of wetlands and fresh water for wildlife. In the Central Valley, a vast interior region that is characterized by a mosaic of wetlands and agricultural lands, we found 22% and 29% decreases in the annual abundance of shorebirds during periods of 3-year drought (2013–2015) and 2-year extreme drought (2014–2015), respectively, when compared to non-drought years. Lower abundance of shorebirds coincided with significant decreases in the mean proportion flooded of survey units (7% and 9%, respectively) that were reliant on fresh water. Drought was associated with lower abundance within both the interior Central Valley and coastal San Francisco Bay for greater and lesser yellowlegs (*Tringa melanoleuca and T. flavipes*) and long- and short-billed dowitchers (*Limnodromus scolopaceus and L. griseus*). Only dunlins (*Calidris alpina*) had patterns of abundance that suggested substantial shifts in distribution between the Central Valley and coastal regions of San Francisco Bay and Point Reyes. Our results indicate that drought has the potential to reduce, at least temporally, shorebird populations and flooded habitat in the Central Valley, and the ability to respond to drought by taking advantage of nearby coastal habitats may limit the long-term effects of drought on some species. Successful conservation strategies must balance the impacts of reduced habitat availability at interior sites with the ability of some migratory shorebirds to adapt rapidly to shifting distributions of resources.

**Funding:** Funding for this research was provided to Point Blue Conservation Science for the Migratory Shorebird Project and Pacific Flyway Shorebird Survey by the David and Lucile Packard Foundation http://packard.org/ and United States Forest Service International Programs https://www.fs. usda.gov/about-agency/international-programs, and to the Migratory Bird Conservation Partnership (Point Blue Conservation Science, Audubon California, The Nature Conservancy) by the S. D. Bechtel, Jr. Foundation http://sdbjrfoundation.org/. The funders had no role in study design, data collection and analysis, decision to publish, or preparation of the manuscript.

**Competing interests:** The authors have declared that no competing interests exist.

# Introduction

Wildlife populations are susceptible to extreme climatic events such as severe drought, and many climate projections suggest that future droughts will likely be longer and drier in many regions [1, 2]. Extreme drought results in freshwater resource scarcity that should be considered when developing effective conservation strategies for both fresh water and wildlife [3, 4]. Evaluating when and where the effects of drought result in resource scarcity, and the resulting influence on wildlife populations, requires long-term monitoring [5]. In the case of highly-mobile species such as migratory birds, long-term monitoring must also occur over a broad extent to document responses to drought among multiple species [6].

Drought can have widespread impacts on avian communities [7, 8], especially those in regions dominated by freshwater ecosystems [9]. Moreover, extensive habitat loss, land modification, and human use of fresh water will compound the effects of drought on migratory birds within interior regions [10]. How avian communities respond to drought will depend on their geographic location, habitat preferences, and the magnitude of reduced habitat availability [11]. For example, waterbirds have been shown tracking habitat availability at a continental scale across a drought cycle in Australia [12], and in southwestern Australia, there was a substantial shift in the amount of suitable habitat available for waterbirds between interior and coastal sites when comparing a wet and dry year [13]. Numerous other studies have demonstrated the vagility of shorebirds by their multi-scale responses to daily [14–16], seasonal [17, 18], and annual [19, 20] variation in resource availability, yet there is a paucity of research on the responses of non-breeding shorebirds to multiyear drought.

Climatic conditions in southwestern North America from 2000 to 2018 have been classified as a megadrought [21] that included the driest four-year period in more than 1200 years in California from 2013 to 2016 [22] and resulted in shortages of fresh water for both humans and wildlife [23, 24]. In California's Central Valley, where < 10% of historic wetlands remain [25], scarcity of fresh water during the drought likely caused food and energy shortfalls for waterfowl [26]. Like waterfowl, nonbreeding shorebird populations rely on managed wetlands across the Central Valley [27], and post-harvest winter-flooded rice fields in the more northerly portions of the region [28]. Non-breeding shorebirds have been shown moving among habitat types [15] and long distances (up to 281 km) within the Central Valley to areas with more stable habitat availability in response to a seasonal reduction in flooded agricultural lands that occurs primarily within rice fields from February to April [29]. Additionally, several studies have shown non-breeding shorebirds moving between habitats in the Central Valley and nearby coastal regions, likely in response to seasonal changes in habitat availability [29–31]. If shorebirds respond similarly over the course of a multiyear drought, there may be long term effects of drought on shorebird abundance and distribution which could limit population sizes in relation to targets for conservation of non-breeding shorebirds in the Central Valley [32]. There is also the potential for continued escalation in competition between humans and wildlife for freshwater resources if a warmer, drier climate persists in the Central Valley [33], which further highlights a need for understanding how shorebirds respond to drought in central California.

In this study, we used annual surveys of non-breeding shorebirds in the Central Valley and nearby coastal sites to characterize the responses of shorebirds to drought, and to document patterns of habitat availability across a drought cycle from 2011 to 2016. Effects of the drought on surface water resources for birds varied among geographic regions and habitat types in the Central Valley, in part because of increasingly widespread curtailments of fresh water for flooding of wetlands and agricultural lands that peaked during 2014 and 2015 [34]. Thus, we used two temporal classifications of drought severity for the multiyear drought, a 3-year

drought from 2013 to 2015 and a 2-year extreme drought from 2014 to 2015 (see Methods for more details), to elucidate the magnitude and timing of effects of drought on shorebirds. Our primary hypothesis was that less surface water (thus lower habitat availability) in the Central Valley during drought years than in non-drought years would lead to lower abundance of all shorebird species, and the magnitudes of decline for both habitat availability (because of weather and water policy changes) and shorebird abundance would be greatest during the 2-year extreme drought period.

We relied on past research to generate predictions for individual shorebird species that would decline in abundance during drought years within managed wetlands or agricultural lands, and those that may shift from the Central Valley to coastal sites. Thus, we predicted lower abundance during drought years than in non-drought years for black-necked stilts (*Himantopus mexicanus*) in managed wetlands, and for killdeers (*Charadrius vociferous*), greater yellowlegs (*Tringa melanoleuca*), and dunlins (*Calidris alpina*) in agricultural lands (primarily rice fields) [35]. Based on the responses of shorebirds to changing distributions of surface water in the Central Valley [29], we also predicted that some shorebirds would shift northward away from the driest, southern areas in the Central Valley which were likely most impacted by drought, and to nearby coastal regions where habitat availability was not dependent on freshwater resources. In the neighboring San Francisco Bay estuary, we expected higher abundance of black-necked stilts [36], dunlins [29], and long-billed dowitchers (*Limnodromus scolopaceus*) [31] during drought years than in non-drought years because they have been documented moving between interior sites and San Francisco Bay. In estuaries of the Point Reyes peninsula, we expected higher abundance of dunlins during drought because of known movements between the Central Valley [30], and for killdeers and least sandpipers (*Calidris minutilla*) because of evidence of higher abundance during dry years in wet years [37–39].

## Study area

Central California includes several regions of importance for shorebirds in the Pacific Flyway [40]. The Central Valley of California is an interior, low elevation basin of approximately 52,000 (km$^2$) between the Sierra Nevada and Coast Ranges and supports over 1 million shorebirds each year [35]. It extends 642 km from north to south and 48 to 112 km from east to west. Runoff from the surrounding montane watersheds is the primary source of fresh water that supports habitat for migratory shorebirds, and water supplies are subject to human management decisions in response to annual and multiannual variation in precipitation. Annual and seasonal precipitation is highly variable ranging from as little as 50% to more than 300% of the long-term averages with most precipitation traditionally occurring from November to March [41]. The climate is progressively drier from north to south, and land use is primarily agriculture with expanding urban areas that are concentrated near the convergence of the Sacramento and San Joaquin rivers. We defined four sub-regions in the Central Valley that were based on modified planning regions of the Central Valley Joint Venture from 2006 [42]: Sacramento Valley, Yolo-Delta, San Joaquin, and Tulare (Fig 1).

Coastal regions included the San Francisco Bay estuary (hereafter San Francisco Bay) and a set of smaller estuaries along the Point Reyes peninsula (Fig 1). San Francisco Bay is the largest estuary on the Pacific coast of North America and centered ~75 km west of the Yolo-Delta sub-region in the Central Valley. San Francisco Bay is used by over half a million shorebirds during migration and winter; see [43] for details on the ecological features and past shorebird use of San Francisco Bay. We consider Suisun Marsh, which lies directly west of the Yolo-Delta sub-region and the convergence of the Sacramento and San Joaquin rivers, as part of San

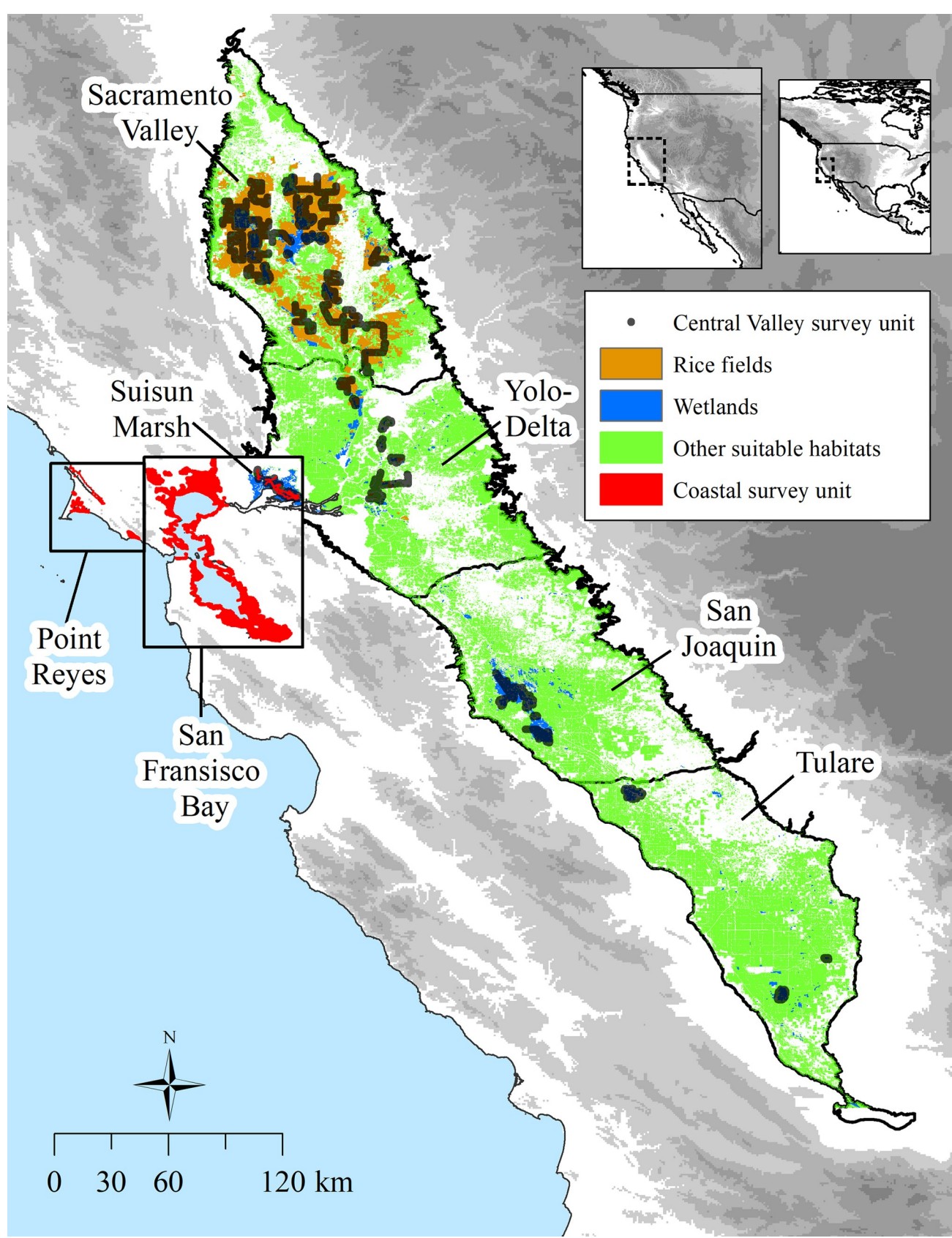

**Fig 1. Study area for annual surveys of non-breeding shorebirds from 2011 to 2016 in California, USA.** Labels are included for Central Valley sub-regions and coastal regions. Shaded relief shows elevation as < 150 m (white) to > 1800 m (darkest gray). Land cover types were derived primarily from satellite imagery; see [29] for more details. See Methods for details on the types of land cover that composed the "other suitable" land cover category. Area of all survey units has been enlarged for the purposes of visualizing the sampling frame.

Francisco Bay region, primarily because the managed wetlands of Suisun Marsh do not rely primarily on deliveries of fresh water for flooding and they lack surrounding agricultural lands, distinguishing it from wetlands in the Central Valley.

The Point Reyes peninsula, on the outer coast ~30 km northwest of San Francisco Bay, includes three smaller, distinct estuaries—Tomales Bay, Bolinas Lagoon, and the Drakes-Limantour estuary system (including Drakes and Limantour Esteros)—and Abbotts Lagoon, a true lagoon. For details on the features and shorebird use of the estuaries of Point Reyes see [36–38].

## Methods

We used data collected annually from 2011 to 2016, for the Pacific Flyway Shorebird Survey (pointblue.org/pfss), a multi-scale survey conducted by biologists and volunteers using a standardized survey protocol and coordinated via a network of partner organizations, which composes part of a network of shorebird monitoring sites along the Pacific Coast of the Americas [6]. Surveys occurred from 15 November to 15 December, when most shorebird migration has ceased and there is controlled flooding of managed wetlands and some post-harvest agricultural lands in the Central Valley [29, 35]. While surveys could occur anytime during the survey period, we scheduled surveys on the same or consecutive days within large complexes of flooded habitats in the Central Valley and within portions of each coastal region unless constrained by access to sites. Surveys were postponed if there were potentially adverse viewing conditions (i.e., weather), constrained access, or limited availability of surveyors. Surveys of coastal sites occurred at a tidal stage that was standardized across years within each estuary; surveys occurred across a high tide in San Francisco Bay, and during a rising tide in each Point Reyes estuary. Each unit was surveyed for a minimum of two minutes or until all individual shorebirds had been counted. To characterize habitat availability, we estimated the proportional area of surface water within each survey unit (hereafter proportion flooded) and recorded the dominant land cover type.

For each defined survey unit, we counted the number of shorebirds (suborders Scolopaci and Charadrii) on the ground [6]. We attempted to identify all individuals to species, but when this was not possible, we used a proportional allocation method that we modified from previous work [40] to correct the totals for species that would have been undercounted because of their inclusion in unidentified, mixed species (UMS) flocks. UMS flocks accounted for 20% of the total candidate species counted in the Central Valley, 10% in San Francisco Bay, and 4% in Point Reyes. In UMS flocks, we counted the total flock size and identified whether at least one of each individual candidate species (only *Calidris* sandpipers in this study) occurred in the flock in the field. We later estimated the proportional composition of candidate species in the flocks based on the composition of those species that were identified in the survey unit, or if the proportion of UMS to total candidate species was greater than one, we used the proportion of UMS to total count of candidate species within the region or sub-region where the flock occurred. Thus, the total for species $i$ was $c_i$, which included the total identified and counted in the field, added to $e_i$, the estimated number included in UMS flocks of size U individuals: $e_i = f_i / (f_i + f_j + ...) * U$, where $(f_i + f_j + ...)$ is the total count of candidate species identified in the field. We allocated counts to western sandpiper (*Calidris mauri*) where

appropriate, but did not analyze data for this species because of substantially lower abundance in the Central Valley when compared to San Francisco Bay [40].

## Survey design

In the Central Valley, we conducted 7,607 surveys of 1,556 individual survey units from 2011 to 2016. We surveyed 58 routes on public and private roads and within complexes of managed wetland impoundments, including 40 routes that were randomly selected in areas with a high potential for use by shorebirds and 18 within wetlands of federal or state refuges [44]. To identify areas with a high potential for use by shorebirds and subsequent random selection of routes, we used a retrospective analysis of satellite imagery of surface water within shorebird habitats and excluded areas that were flooded from November to January in < 30% of years 2000 to 2010, which removed approximately 90% of the Central Valley. Forty township grids (9.66 x 9.66 km; Public Land Survey System; http://www.nationalatlas.gov) were then randomly selected to contain a single survey route [45]. Overall, grids selected for survey routes in the Sacramento Valley and Yolo-Delta sub-regions had ≥ 50% of 30 x 30 m pixels classified as rice fields ($n$ = 35 grids). Grids selected in the San Joaquin sub-region had ≥ 50% of cells classified as managed wetlands ($n$ = 5). The remaining 18 routes were within federal or state refuges from the Sacramento National Wildlife Refuge (NWR) Complex in the Sacramento Valley sub-region south to Kern and Pixley NWRs in the Tulare sub-region, except for the Cosumnes River preserve managed by the Nature Conservancy in the Yolo-Delta sub-region. Each randomly selected route consisted of 14–20 survey units. In total, we surveyed 1,856 ± 146 km$^2$ annually in the Central Valley; the Sacramento Valley accounted for 80%, Tulare 8%, San Joaquin 7%, and Yolo-Delta 7% (S1 Table).

In the Central Valley, we used three types of fixed spatial boundaries to delineate survey units: fixed-radius point counts ($n$ = 774), restricted area searches ($n$ = 409), and complete area searches ($n$ = 377). Fixed radius (0.16 km) point counts were used only for randomly-selected road routes and spaced approximately every 0.8 km along routes that were up to 23 km in length on publicly accessible roads. We used restricted area searches for seasonally flooded impoundments on federal, state, or private lands; survey unit boundaries were located a fixed distance (0.16 km) from roads and bounded on the sides by levees that defined each wetland management unit. We used complete area searches for comprehensive counts of individual seasonally flooded impoundments within federal refuges; survey unit boundaries were defined entirely by levee boundaries of individual impoundments, which varied in size. Complete area searches were conducted by U.S. Fish and Wildlife Service personnel exclusively within the Sacramento NWR and Kern-Pixley NWR complexes.

In coastal regions, we surveyed primarily within tidally-influenced wetlands, intertidal mudflats, tidal salt marsh, managed ponds, salt production ponds and other associated or nearby managed impoundments. We designed boundaries of coastal survey units using coastline boundaries, habitat delineations, levees, and a maximum distance from the coastline of 300 m. In San Francisco Bay, we selected survey units using a stratified random process that weighted units using the natural logarithm of total shorebirds counted during comprehensive surveys from 2006 to 2008 [46]. This resulted in 1,095 surveys ($n$ = 258 survey units) with an annual mean area of 1,571± 61 km$^2$ surveyed (S1 Table); there was no survey in Suisun Marsh during 2011 and some portions of San Francisco Bay were not accessible in 2014. In Point Reyes, we conducted comprehensive surveys of each estuary using boundaries that aligned with pre-existing survey units in Bolinas Lagoon [39], Tomales Bay [38], and Limantour Estero, an arm of Drakes Estero [37]. In 2013, we added additional survey units for Abbotts Lagoon and intertidal habitats within the remainder of Drakes Estero. This resulted in 330 surveys of 64 survey units with an annual mean of 149 ± 25 km$^2$ surveyed in Point Reyes.

## Dominant land cover types in the Central Valley

We categorized individual survey units in the Central Valley into four land cover types: managed wetlands (58% of total area surveyed annually), rice fields (19%), other suitable habitats (21%), and miscellaneous (2%). Managed wetlands (wetlands) and post-harvest rice fields (rice fields) are important targets for conservation actions and account for the majority of potential shorebird habitat that is consistently flooded at depths and durations that benefit shorebirds in the Central Valley [32]. We defined "other suitable habitats" as habitats other than wetlands and rice fields that had high potential for use by shorebirds when flooded, including grasslands/pasture, corn fields, other irrigated row or field crops, freshwater lakes/ponds, and wastewater treatment ponds. We did not sample any individual other suitable habitat extensively. We defined "miscellaneous" land cover types as having low potential for use by shorebirds, and included developed, forested, riverine, orchard, and vineyard.

We identified a single dominant cover type for each survey unit using a combination of satellite imagery and data collected during field surveys. We began by using a spatial data layer depicting land cover in the Central Valley (30 x 30 m pixels) [29]; the dominant cover type had the highest proportion of land cover within each survey unit boundary. We used the estimated cover type from GIS data when it was consistent with field observations or when the latter were not available. When the GIS-estimated cover type differed from the observed cover type, we manually assigned a cover type by visually assessing satellite imagery of the unit from recent years in Google Earth Engine (explorer.earthengine.google.com). Cover types for individual units remained static across years except for eight units for which both an observer and satellite imagery documented a change in the dominant land cover type.

## Focal species

In the Central Valley, we investigated the effects of drought on the abundance of all shorebirds species combined and six focal species. In coastal regions, we investigated only focal species because the geographic distribution of some shorebird species is primarily on the coast (e.g., western sandpiper). Focal species commonly overwintered within both interior and coastal wetlands of central California; we identified killdeers, black-necked stilts, greater yellowlegs, dunlins, least sandpipers, and long-billed dowitchers as focal species because of their relative abundance and occurrence during November and December in both the Central Valley and coastal California [40]. We combined counts of greater yellowlegs and lesser yellowlegs (*Tringa flavipes*; hereafter yellowlegs) as well as long-billed dowitchers and short-billed dowitchers (*Limnodromus griseus*; hereafter dowitchers) because they often could not be distinguished during surveys. However, for yellowlegs and dowitchers, we considered our findings applicable only to the greater yellowlegs and long-billed dowitcher, respectively, because past research found that they composed nearly all of their species grouping that was encountered during November in the Central Valley [35]. Moreover, long-billed dowitchers have been shown to account for nearly all dowitchers in San Francisco Bay during the wintering period [47]. Other species of shorebirds that we encountered in the Central Valley included: black-bellied plover (*Pluvialis squatarola*), snowy plover (*Charadrius nivosus*), American avocet (*Recurvirostra americana*), long-billed curlew (*Numenius americanus*), willet (*Tringa semipalmata*), marbled godwit (*Limosa fedoa*), western sandpiper, Wilson's phalarope (*Phalaropus tricolor*), and Wilson's snipe (*Gallinago delicata*).

## Data summary and analysis

We analyzed 8,979 surveys of shorebirds using counts that were corrected for differences in the area surveyed (S1 Dataset), and we considered the resulting model-based estimates of

density of shorebirds (count/survey area in km$^2$) to be a proxy for abundance. See S1–S5 Tables for data summaries that include mean annual counts and annual density estimates of focal species by region, sub-region, and land cover type. In the Central Valley, we summed counts of all shorebird species (hereafter total shorebirds) and grouped total shorebird counts by sub-region and land cover type; whereas for counts of individual focal species and estimates of the proportion flooded of survey units, we grouped observations by land cover type only. In San Francisco Bay, we analyzed counts of focal species and estimates of the proportion flooded of survey units. In Point Reyes, we analyzed counts of focal species, and we did not analyze data on flooding of survey units because of a paucity of data at most sites during 2011 and 2012. To qualitatively assess shifts in patterns of abundance of shorebirds within the Central Valley, we calculated a standardized density for each sub-region as the mean annual density of total shorebirds from 2011 to 2016, and then mapped the annual density of total shorebirds relative to the standardized mean by sub-region.

We used generalized linear mixed effects models to test for the effects of drought on both annual abundance and habitat availability (proportion flooded of survey units). Models tested for differences in a binary factor that characterized each survey as occurring during a drought (1) or a non-drought year (0). We identified 2013 to 2015 as drought years based on the water year type as defined by the California Department of Water Resources for the Sacramento and San Joaquin River Valleys based on the projected runoff (million acre feet) on 1 May (see http://cdec.water.ca.gov/cgi-progs/iodir/WSIHIST for details on the level for each classification and data access) [24]. Over the six years of the study, the Sacramento Valley was classified as drought or critical from 2013 to 2015, below normal (2012, 2016) and wet (2011). The San Joaquin Valley (San Joaquin and Tulare sub-regions) was classified as drought or critical in all years except 2011. For the entire Central Valley, we considered a drought year to include any water year designated as "drought" or "critical" in both the Sacramento and San Joaquin Valleys. We then analyzed both 3-year drought (2013–2015) and 2-year extreme drought (2014–2015) periods separately; we defined an extreme drought year as having a drought year immediately preceding it.

The model structure for predicting shorebird abundance ($y$) was: $y = \exp(\beta_{0i} + \beta_{1i}X_i + \text{offset}(A_j) + b_j + e_k)$, where $y$ was the shorebird count, $\beta_0$ and $\beta_1$ was intercept and slope of the specified metric for data grouping $i$, $X$ was a binary metric for drought year at survey unit $j$, $A$ was the natural log of area surveyed (km$^2$), $b$ was a random effect term to account for repeated surveys at individual survey units, and $e_k$ was a parameter to capture overdispersion. For models of habitat availability, we used the same model structure except that $y$ represented the proportion flooded of survey units and the offset term for area surveyed was excluded. Because we considered the proportion flooded to be normally distributed, parameter estimates did not require back-transformation. We fitted all models using maximum likelihood estimation and we used a negative binomial distribution with a "log" link for models of shorebird abundance. For negative binomial models, we report fixed factor coefficients ($\beta_1$) with 95% confidence intervals after back transforming them (with the inverse of the link function); thus reported $\beta_1$ represent the proportional change in shorebird abundance from non-drought to drought years. To test for significance, we used a null-hypothesis likelihood ratio test on the $t$-statistic for each parameter; we considered $p \leq 0.05$ to represent moderate (0.01–0.05) to convincing ($< 0.01$) evidence for a difference in response variables between non-drought and drought years, and $p = 0.06$ to 0.10 to provide suggestive but inconclusive evidence of such a difference [48, 49]. We compared among models of the same response using deviance; models with lower deviance values represent relatively better fit to the data. All analyses were conducted in Program R (R Core Team, Version 3.5.1; http://www.r-project.org/) and package glmmTMB [50].

## Results

### Patterns of shorebird use in the interior

Model estimates of total shorebird abundance across all cover types in the Central Valley provided moderate to convincing evidence of lower abundance during drought years, including a 22% and 29% declines during the periods of 3-year drought (β = 0.78, 95% CI: 0.63, 0.98, $p$ = 0.03) and 2-year extreme drought (β = 0.71, 95% CI: 0.56, 0.91, $p$ = 0.007; Fig 2), respectively, when compared to non-drought years. There was modest increase in model fit from the period of 3-year drought (deviance = 22,886.4) to 2-year extreme drought (22,883.8) for models of total shorebird abundance across the Central Valley. Among models of total shorebirds by cover type, we found no significant differences in abundance during the periods of 3-year drought or 2-year extreme drought than in non-drought years within wetlands (β = 0.89, 95% CI: 0.65, 1.23, $p$ = 0.14; β = 0.78, 95% CI: 0.54, 1.11, $p$ = 0.17), rice fields (β = 0.78, 95% CI: 0.55, 1.09, $p$ = 0.49; β = 0.75, 95% CI: 0.52, 1.09, $p$ = 0.14), or other suitable habitats (β = 0.74, 95% CI: 0.32, 1.68, $p$ = 0.47; β = 0.64, 95% CI: 0.26, 1.60, $p$ = 0.34); however, in miscellaneous

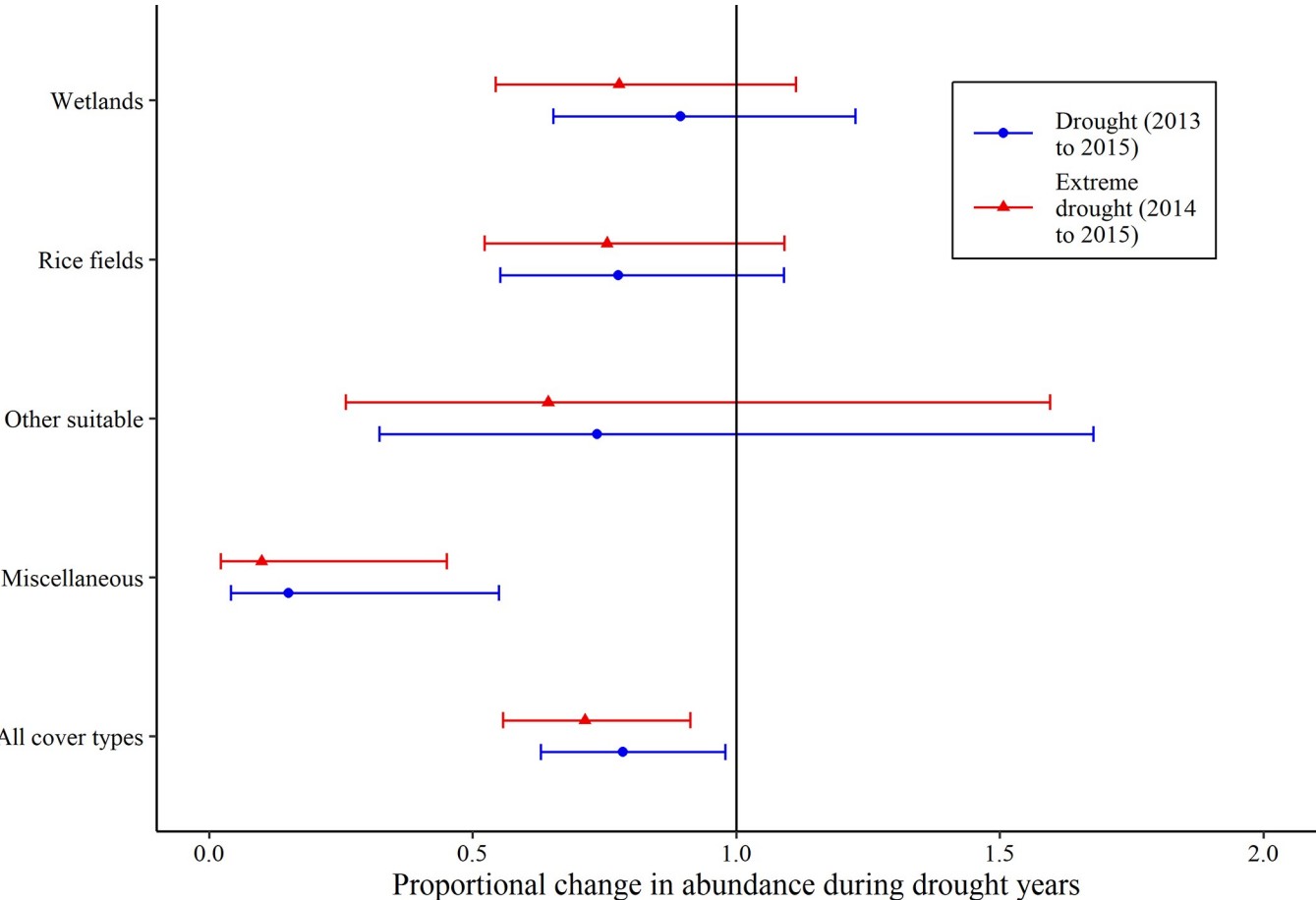

**Fig 2. Estimated abundance of total shorebirds during a drought year among four dominant cover types.** Results from generalized linear mixed models of the annual abundance of total shorebirds from 2011 to 2016 in the Central Valley, California, USA. Plots for each land cover type show the estimated proportional change in total shorebird abundance relative to a non-drought year from two models; drought models classified drought years as a period of a 3-year drought (2013–2015) or 2-year extreme drought (2014–2015). The solid black line represents no change from the estimated abundance of total shorebirds during a non-drought year. See Methods for details on the types of land cover that composed the "other suitable" and "miscellaneous" land cover categories.

cover types, we found convincing evidence of lower abundance of total shorebirds during both the periods of 3-year drought (β = 0.15, 95% CI: 0.04, 0.55, $p$ = 0.004) and 2-year extreme drought (β = 0.10, 95% CI: 0.02, 0.45, $p$ = 0.003) than in non-drought years. There were modest increases in model fit from the period of 3-year drought to 2-year extreme drought, for models of total shorebird abundance in wetlands (deviance = 10959.6 and 10958.2, respectively), other suitable (1848.6 and 1848.2, respectively), and miscellaneous habitats (456.4 and 455.9, respectively), but no difference in rice fields (both 9403.8).

Among models of focal species, we found lower abundance of killdeers across all land cover types during both the periods of 3-year drought and 2-year extreme drought than in non-drought years and for dowitchers during the period of 3-year drought only (Fig 3 and S2 Table). For models of abundance in wetlands, we found moderate evidence of lower abundance in both the periods of 3-year drought and 2-year extreme drought than in non-drought years for black-necked stilts (32% and 38% declines, respectively; Table 1), suggestive evidence of lower abundance during the 2-year extreme drought for least sandpipers (51% decline), and moderate evidence of lower abundance over the period of 3-year drought for dowitchers (46% decline). In rice fields, we found moderate evidence of lower abundance during both the

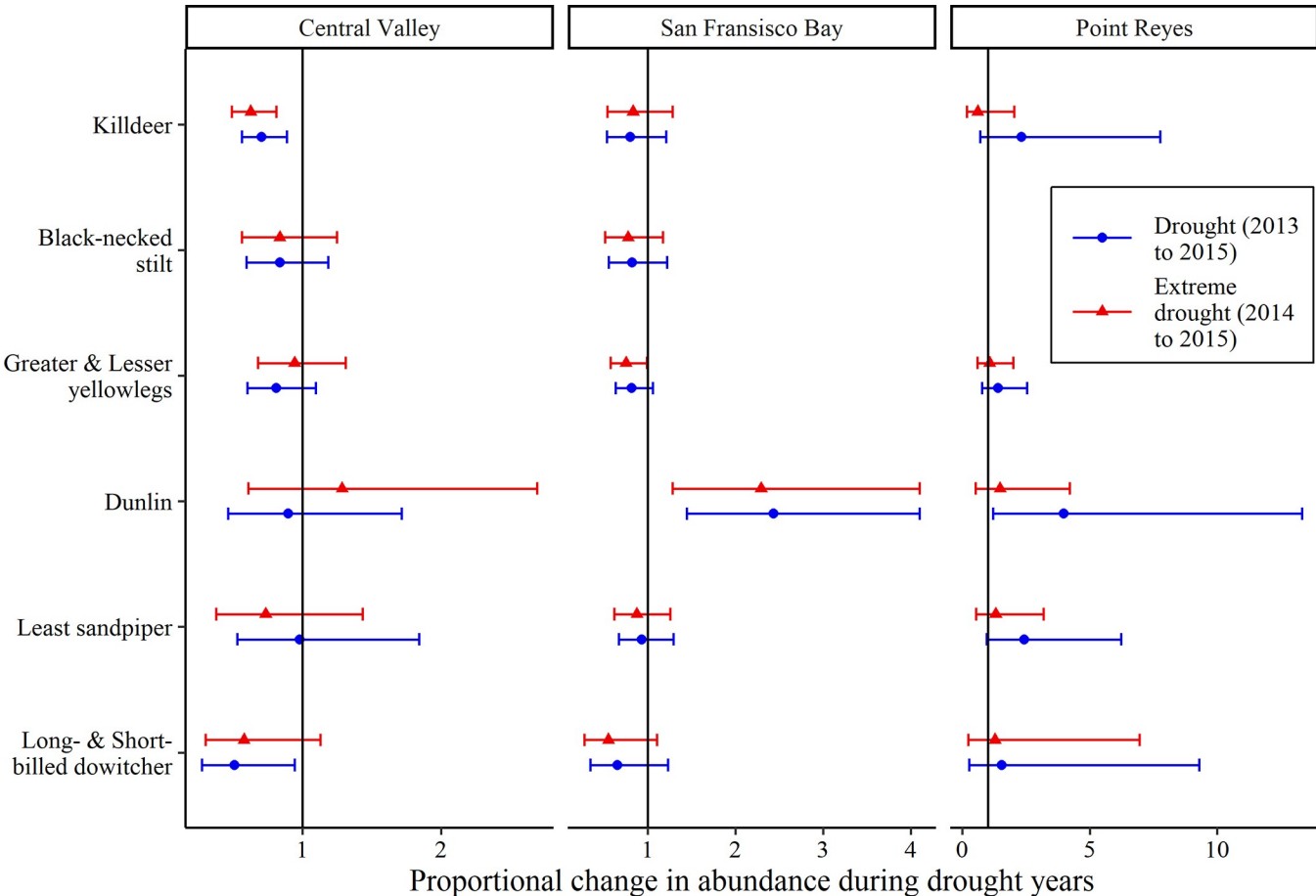

**Fig 3. Estimated abundance for six focal species during a drought year in central California, USA.** Results from generalized linear mixed models of the annual abundance of six shorebirds from 2011 to 2016 in three regions of central California, USA. Plots for each region show the estimated proportional change in total shorebird abundance relative to a non-drought year from two models; drought models classified drought years as a period of a 3-year drought (2013–2015) or 2-year extreme drought (2014–2015). The solid black line represents no change from the estimated abundance of total shorebirds during a non-drought year.

**Table 1. Results of models of annual shorebird abundance among cover types in the Central Valley, California, USA.**

| Focal species | Model | Wetlands | | | | Rice fields | | | | Other suitable habitats | | | |
|---|---|---|---|---|---|---|---|---|---|---|---|---|---|
| | | Deviance | β | 95% CI | | p-value | Deviance | β | 95% CI | | p-value | Deviance | β | 95% CI | | p-value |

| Focal species | Model | Deviance | β | 95% CI | | p-value | Deviance | β | 95% CI | | p-value | Deviance | β | 95% CI | | p-value |
|---|---|---|---|---|---|---|---|---|---|---|---|---|---|---|---|---|
| Killdeer | Drought | 3,443.6 | 1.05 | 0.74 | 1.48 | 0.79 | 5,092.3 | 0.69 | 0.33 | 1.41 | *0.03* | 1,048.4 | 0.55 | 0.23 | 1.35 | 0.20 |
| | Extreme drought | 3,443.5 | 0.83 | 0.54 | 1.26 | 0.38 | 5,090.4 | 0.67 | 0.49 | 0.91 | *0.01* | 1,048.8 | 0.56 | 0.21 | 1.51 | 0.25 |
| Black-necked stilt | Drought | 6,998.6 | 0.68 | 0.48 | 0.98 | *0.04* | 769.7 | 1.26 | 0.33 | 4.86 | 0.74 | 401.4 | 0.90 | 0.15 | 5.36 | 0.91 |
| | Extreme drought | 6,997.9 | 0.62 | 0.41 | 0.95 | *0.03* | 769.8 | 0.96 | 0.22 | 4.12 | 0.95 | 401.4 | 0.98 | 0.13 | 7.60 | 0.98 |
| Greater & Lesser yellowlegs | Drought | 2,934.4 | 0.89 | 0.60 | 1.30 | 0.54 | 2,374.5 | 0.77 | 0.47 | 1.25 | 0.29 | 285.8 | 2.36 | 0.49 | 11.31 | 0.29 |
| | Extreme drought | 2,934.3 | 1.15 | 0.76 | 1.74 | 0.51 | 2,372.8 | 0.63 | 0.36 | 1.07 | 0.09 | 327.1 | 1.54 | 0.30 | 7.99 | 0.61 |
| Dunlin | Drought | 2,174.1 | 0.78 | 0.34 | 1.77 | 0.55 | 2,550.9 | 1.02 | 0.44 | 2.34 | 0.97 | 222.9 | <0.001 | <0.001 | 0.045 | *0.003* |
| | Extreme drought | 2,174.0 | 0.72 | 0.30 | 1.76 | 0.47 | 2,550.0 | 1.53 | 0.63 | 3.73 | 0.35 | 223.5 | 0.27 | 0.01 | 12.81 | 0.51 |
| Least sandpiper | Drought | 2,619.9 | 0.74 | 0.37 | 1.46 | 0.38 | 2,443.1 | 1.06 | 0.41 | 2.73 | 0.91 | 355.0 | 0.001 | <0.001 | 0.059 | *0.002* |
| | Extreme drought | 2,617.5 | 0.49 | 0.23 | 1.03 | *0.06* | 2,443.1 | 0.95 | 0.34 | 2.62 | 0.92 | 345.5 | 2.70 | 0.16 | 46.17 | 0.49 |
| Long- & Short-billed dowitcher | Drought | 3,670.8 | 0.54 | 0.28 | 1.05 | *0.07* | 1,872.4 | 0.32 | 0.08 | 1.19 | 0.09 | 291.3 | 13.44 | 1.15 | 157.59 | *0.04* |
| | Extreme drought | 3,673.1 | 0.69 | 0.33 | 1.41 | 0.31 | 1,873.4 | 0.40 | 0.12 | 1.40 | 0.15 | 294.0 | 1.78 | 0.03 | 124.84 | 0.79 |

periods of 3-year drought and 2-year extreme drought than in non-drought years for killdeers (31% and 33% declines, respectively), suggestive evidence of lower abundance during the period of 2-year extreme drought for yellowlegs (37% decline), and suggestive evidence of lower abundance during the period of 3-year drought for dowitchers (62% decline). Within other suitable habitats, we found convincing evidence of lower abundance of dunlins and least sandpipers during the period of 3-year drought than in non-drought years, and the size of the estimated effect showed abundance during the period of 3-year drought was less than 1% of non-drought abundance for both species. We found moderate evidence for an effect of 3-year drought on the abundance of dowitchers in other suitable habitats, but in contrast to our prediction, they increased in abundance during drought years compared to non-drought years.

Results of generalized linear mixed models of abundance of six focal shorebirds in three dominant land cover types used by shorebirds in the Central Valley. Data collected during annual shorebird surveys from 2011 to 2016. Models tested differences in abundance between non-drought years and drought years (2013 to 2015 for drought; 2014 to 2015 for extreme drought). Parameter estimates (β) and 95% confidence intervals (CI) are back-transformed to the scale of the response variable and represent the proportional change in shorebird abundance from non-drought to drought years. *P*-values are from a null-hypothesis likelihood-ratio test on the *t*-statistic for each parameter. Statistically significant *p*-values are in italics. Deviance scores estimated the relative goodness of fit for each model for a specific species; a model with a lower score represents better goodness of fit. See Methods for details on the types of land cover that composed the "other suitable" land cover category.

Geographic patterns of the relative mean density of total shorebirds varied annually and among sub-regions in the Central Valley (Fig 4; S1 Fig). Across all sub-regions in the Central Valley, shorebird density was highest during 2012 (0.0012 birds per $km^2$), and more than twice the lowest density observed during 2015 (0.0005; S1 Table). In 2012, we also documented maximum annual estimates of abundance for black-necked stilts, dowitchers, and five of eight non-focal species detected in the Central Valley. In 2014, higher than average shorebird

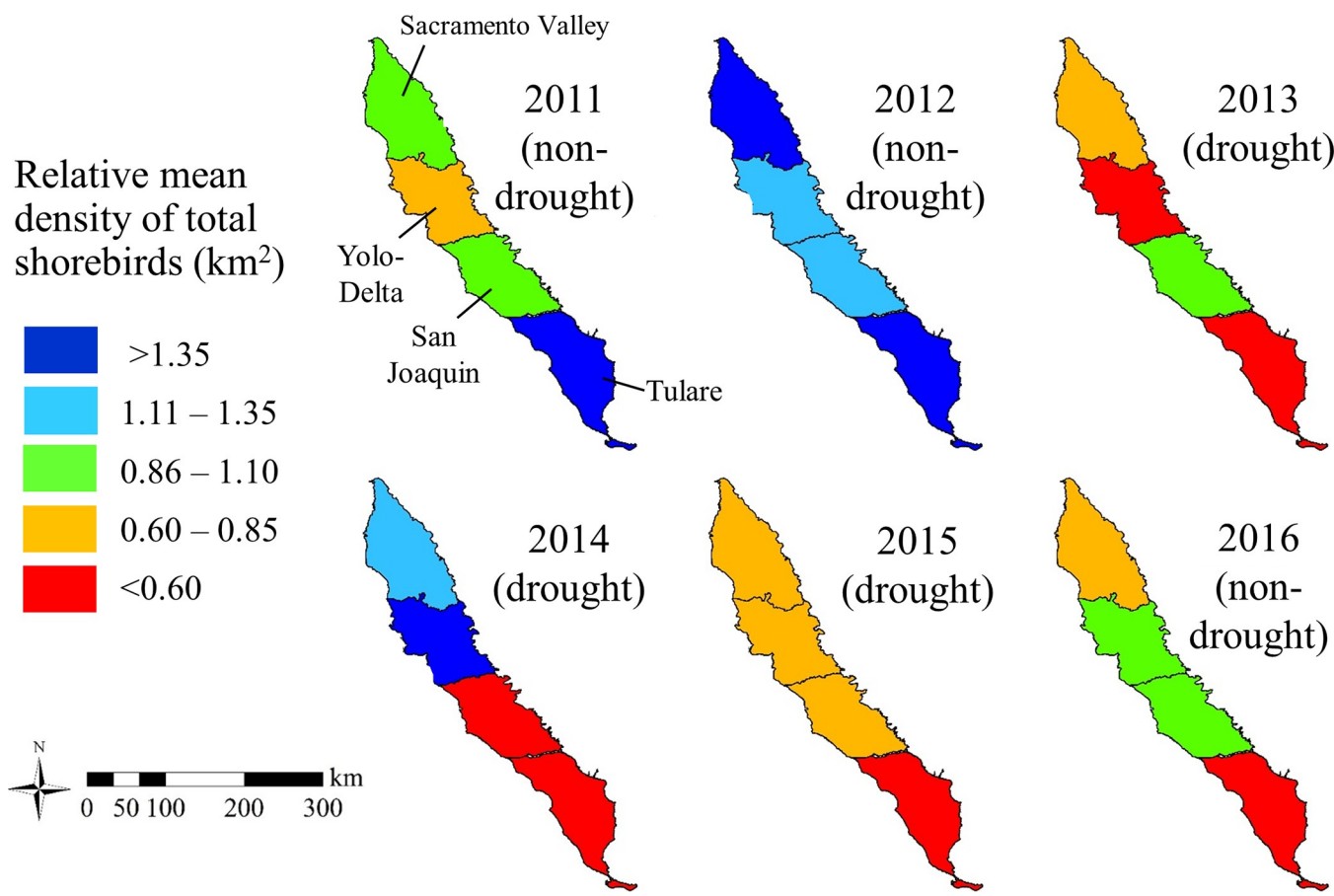

**Fig 4. Relative mean density of total shorebirds (birds per km²) by sub-region from 2011 to 2016 in the Central Valley, California, USA.** The density categories show the proportional difference between the annual mean estimate of density and the standardized mean density from 2011 to 2016 in each sub-region; a relative mean density of 1.0 is equal to the standardized mean annual density (birds per km²).

density in northern sub-regions was likely related to three of the five highest counts of total shorebirds, which consisted primarily of dunlins, occurring in survey units of rice fields near the Sacramento River. Of the twenty highest counts of total shorebirds, 5% occurred in 2013, 25% in 2014 (all near the Sacramento River), and 5% in 2015. The density of shorebirds was considerably higher than average during 2012 and 2014, and lower than average during 2013 and 2015 in both the Sacramento Valley and Yolo-Delta sub-regions. San Joaquin was the only sub-region where annual density of total shorebirds was near average in 2013; no survey occurred at the Los Banos Wildlife Area in 2014, which may have contributed to the low relative mean density in this sub-region. In the Tulare sub-region, annual density declined rapidly from 2012 to 2013 and then remained low. Notably, no sub-region had higher than average shorebird density in the non-drought year of 2016.

Despite wetlands accounting for the majority of area surveyed in the Central Valley (58%; S3 Table), only two of six focal species had a majority of individuals counted within wetlands. For killdeers, 59% of 4,994 individuals were in rice fields, 24% in wetlands, 12% in other suitable, and 5% in miscellaneous habitats; killdeers also accounted for 80% of the 355 individual shorebirds counted within miscellaneous. For black-necked stilts, 89% of 11,982 individuals were in wetlands, 6% in rice fields, 5% in other suitable, and 0% in miscellaneous habitats. For yellowlegs, 52% of 2,387 individuals were in wetlands, 43% in rice fields, 4% in other suitable,

and < 1% in miscellaneous habitats. For dunlins, 71% of 20,502 individuals were in rice fields, 26% in wetlands, 3% in other suitable, and 0% in miscellaneous habitats. For least sandpipers, 55% of 16,724 individuals were in rice fields, 36% in wetlands, 9% in other suitable, and < 1% in miscellaneous habitats. For dowitchers, 55% of 28,735 individuals were in rice fields, 41% in wetlands, 5% in other suitable, and < 1% in miscellaneous habitats (S4 Table). We counted comparatively few individuals of uncommon to rare non-focal shorebird species, particularly during 2015 and 2016 (S5 Table).

## Coastal abundance

We found convincing evidence for higher abundance of dunlins in San Francisco Bay during the periods of 3-year drought (143% increase; $p < 0.001$) and 2-year extreme drought (130%; $p = 0.005$) than in non-drought years, and moderate evidence of higher abundance of dunlins in Point Reyes during only the period of 3-year drought (299% increase; $p = 0.02$; Fig 4 and S1 Table). We also found suggestive evidence for an increase in abundance of least sandpipers during the 3-year period of drought when compared to non-drought years in Point Reyes (143% increase; $p = 0.06$). For other focal species, models of abundance in San Francisco Bay indicated no difference or lower abundance during drought years than in non-drought years, and only no difference in Point Reyes. Lower abundance in San Francisco Bay during the 2-year period of extreme drought than in non-drought years was contrary to our predictions for yellowlegs (25% decline; $p = 0.04$) and dowitchers (44%; $p = 0.095$). Annual density estimates of focal species peaked from 2013 to 2016 in San Francisco Bay, including maximum annual estimates for dowitchers in 2013, black-necked stilts in 2014, killdeers and dunlins in 2015, and yellowlegs in 2016 (S6 Table). In Point Reyes, we recorded zero detections of black-necked stilts and few dowitchers when compared to other regions.

## Habitat availability

Models of habitat availability in the Central Valley provided convincing evidence of less flooding of survey units during both the periods of 3-year drought and 2-year extreme drought when compared to non-drought years, including 7% and 9% declines, respectively (Fig 5; S7 Table). Among cover types, we found convincing evidence of less flooding of survey units during both the periods of 3-year drought and 2-year extreme drought within wetlands (7% and 9% declines, respectively), rice fields (10% and 11% declines), and other suitable habitats (5% and 7% declines; all $p < 0.001$). There were no differences in the proportion flooded of survey units for miscellaneous habitats (both periods of drought $p > 0.78$). In San Francisco Bay, models found no difference in the proportion flooded of survey units during the period of 3-year drought when compared to non-drought years, and a moderate effect of increased flooding during the period of 2-year extreme drought ($p = 0.02$), which was possibly related to habitat restoration in some units and variations in the maximum tidal height during the survey each year.

In the Central Valley, the mean proportion flooded of survey units was highest in 2012 when approximately 942 of 1,744 $km^2$ (0.54) surveyed was flooded, and then the annual mean declined until the lowest estimate in 2015 when approximately 684 of 1,753 $km^2$ (0.39) surveyed was flooded. The mean proportion flooded of survey units during the post-drought year in 2016 was similar to 2011 (0.48). Annual estimates of the mean proportion flooded of survey units by sub-region indicated that the Yolo-Delta and San Joaquin sub-regions had less variable habitat availability from 2011 to 2016 than the Sacramento Valley and Tulare sub-regions (S2 Fig).

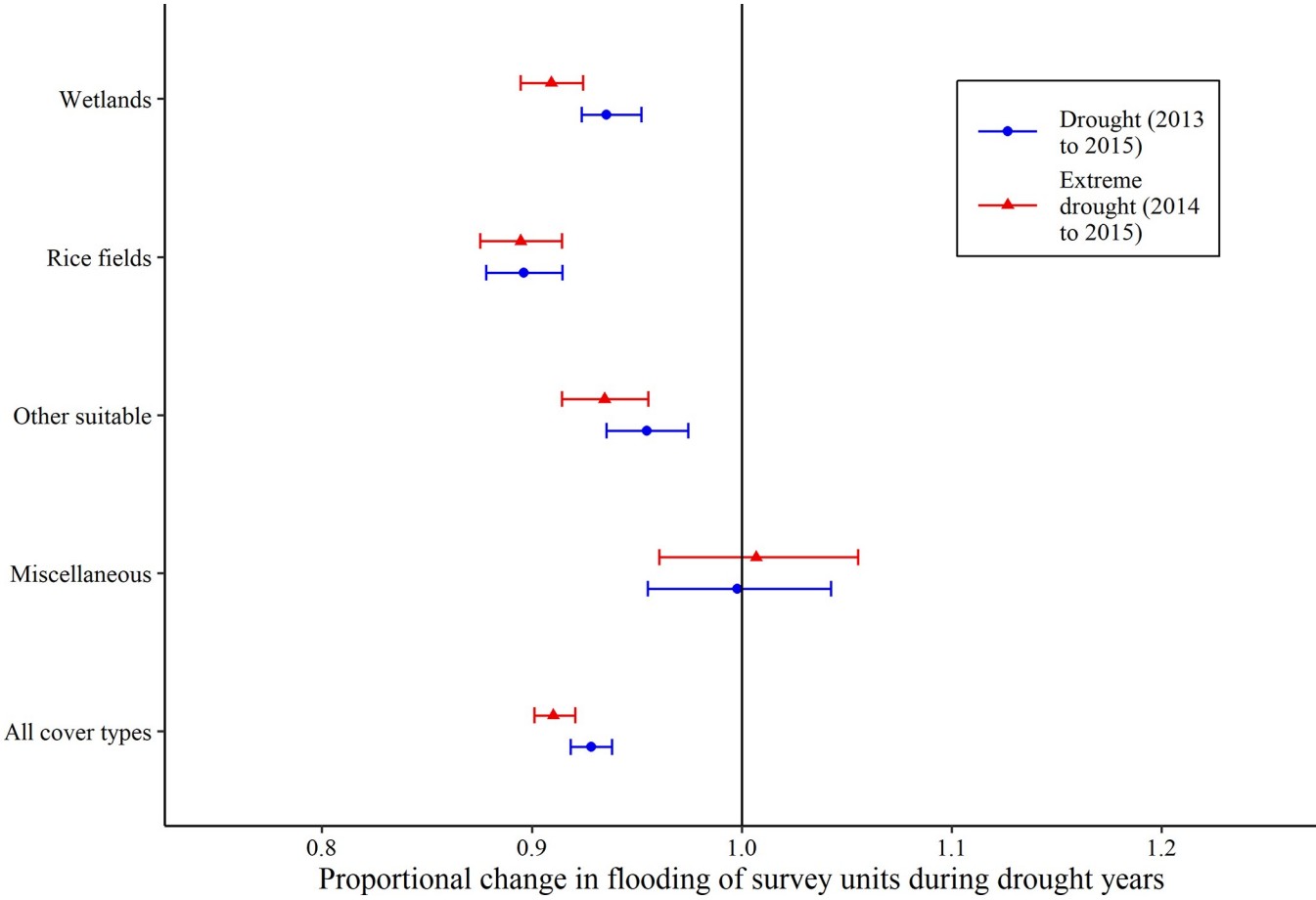

**Fig 5. Estimated proportional change in flooding of survey units during drought years.** Results from generalized linear mixed models of the proportion flooded of shorebird survey units from 2011 to 2016 in the Central Valley California, USA. Plots for each land cover type show the estimated change in the annual proportion flooded of survey units from the 3-year drought (2013–2015) model or 2-year extreme drought (2014–2015) model. The solid black line represents the proportion flooded of survey units during a non-drought year. See Methods for details on the types of land cover that composed the "other suitable" and "miscellaneous" land cover categories.

## Discussion

Our study revealed that drought lowered shorebird abundance and habitat availability in the Central Valley, including evidence that the magnitude of decline increased as the drought became more extreme. There was mixed support for hypotheses of lower abundance during drought years at the focal species and land cover type levels, and only dunlin and least sandpiper had patterns of abundance that suggested shifts in distribution among Central Valley and coastal habitats. Notably, both yellowlegs and dowitchers had lower abundance during drought years within both an interior and coastal region which suggests adverse effects of the drought on their populations across central California. Our only finding that contradicted hypotheses related to a species' historically preferred land cover type was an unexpectedly high abundance of dunlins in rice fields during some drought years, which was likely related to dunlins congregating within remaining flooded habitats in the Sacramento Valley and Yolo-Delta sub-regions as the drought progressed. Patterns of abundance were mostly similar to patterns of habitat availability across the Central Valley, indicating a possible density-dependent response by shorebirds to reduced habitat availability over the course of this multiyear drought [51].

During this 6-year study, the years prior to and immediately after the 3-year drought period from 2013 to 2015 had notable patterns of shorebird abundance and habitat availability. In 2012, annual shorebird abundance peaked in response to widespread flooding of shorebird habitats that was supplemented by seasonal precipitation before and during the survey period. Prior to winter 2012–2013, four of six previous winters had less than average precipitation and there was already a declining trend in the habitat availability and abundance of colonial breeding herons and egrets (family Ardeidae) in the Central Valley [52]. In 2016, which was the first year of average precipitation after the most extreme years of drought in 2014 and 2015, our findings suggested a potential time lag in the response of non-breeding shorebirds repopulating some flooded habitats [7, 11]. In 2016, patterns in habitat availability and total shorebird abundance diverged in the Sacramento Valley and Tulare sub-regions, where the magnitude of decline in flooding was most substantial during drought years, with proportionally fewer shorebirds in those regions than the amount of flooding might suggest when compared to the pre-drought years of 2011 and 2012. Potential factors that may have contributed to a time-lagged response in repopulating these sub-regions included younger generations of shorebirds having less familiarity with these areas and because sufficient amounts of invertebrate prey may not have reestablished in areas that were not flooded during one or more drought year [19].

Interspecies variation in responses to drought can be better understood when considering differences in the ecological tendencies of focal species. The killdeer and black-necked stilt were the only focal species whose annual breeding productivity may have been directly affected by the drought, as both species breed and winter within central California. It's reasonable to consider that lower breeding productivity during drought years for killdeers and black-necked stilts contributed to lower abundance during early winter in the Central Valley, especially when also considering that breeding populations of both killdeers and black-necked stilts declined in the Central Valley from 2004 to 2013 [53]. Moreover, extremely dry climatic conditions were associated with lower productivity of colonial breeding waterbirds from 1991 to 2010 in San Francisco Bay [54]. Long-term population trends for wintering bird populations in the Central Valley from 1978–1979 to 2013–2014 included a substantial decline of killdeers and a substantial increase of black-necked stilts [55]. Although we did not document patterns of abundance of killdeers that suggested shifting distributions, another study found that some killdeers relied on a natural freshwater wetland as drought refugia. Killdeers had a higher density from 2013 to 2015 in proximity of a relatively small, natural akali wetland complex in the Carrizo Plain, an interior basin approximately 80 km west of Kern NWR in the Tulare sub-region [5]. We also considered that if breeding populations of both killdeers and black-necked stilts were in decline during our study, our ability to detect patterns of abundance that suggested shifts in distribution may have been reduced. In addition, black-necked stilts were wetland specialists compared to other focal species and their results highlight the effects of drought on interior managed wetlands in the Central Valley. For yellowlegs, we found evidence that drought lowered abundance during only the period of 2-year extreme drought, suggesting a time-lagged response that occurred between the extreme drought years of 2014, when their annual abundance peaked across the Central Valley, and 2015 when their annual abundance was lowest.

Shifts in distribution were most substantial for populations of dunlins, and to a lesser extent least sandpipers. As the drought intensified in the Central Valley during 2013 and 2014, large flocks of *Calidris* sandpipers, mostly composed of dunlins, were observed in remaining flooded habitats in the Sacramento Valley and Yolo-Delta sub-regions and to nearby coastal regions. Large aggregations of shorebirds in 2014 near the Yolo bypass were notable because our estimate of habitat availability in the Yolo-Delta sub-region was the lowest in 2014, and the

complex of wetlands and flooded agricultural lands that comprises the Yolo bypass is a geographic bottleneck that serves as a movement corridor for wintering waterbirds between the Sacramento Valley and Yolo-Delta sub-regions [29, 56]. Our observations during 2014 suggested that the costs of increased competition for resources resulting from larger aggregations were still lower than or equal to those associated with moving to the coast, and that in 2015, when the amount of surface water flooding wetlands and agricultural lands reached minimum levels in the Central Valley [34], a threshold was passed whereby those relative costs switched and some shorebirds (dunlins, least sandpipers) moved to the coast. Shifting patterns of abundance by dunlins in our study indicate a high level of mixing of regional wintering populations that has likely contributed to genetic homogeneity among dunlins overwintering along the Pacific coast of North America [57]. There may also be costs when utilizing unfamiliar or less preferred habitat and food resources, such as shifting from interior flooded rice fields to coastal wetlands, including more niche overlap with other shorebirds [58], and potentially an increased risk of predation [59].

Least sandpipers have similar flocking and foraging tendencies to dunlins, yet their distribution shifts in response to drought were less substantial. Both *Calidris* sandpipers forage primarily by probing in mud around or within shallow water, and the amount and resources available within these microhabitats may have been less impacted by drought than the deeper waters preferred by black-necked stilts and yellowlegs that forage high in the water column or the drier terrestrial habitats near water preferred by foraging killdeer. Our study highlighted the importance of flooded rice fields to least sandpipers, given they had lower abundance during drought in wetlands and other suitable habitats in the Central Valley, and supported previous studies that found higher abundance of least sandpipers (and most other focal species) during dry years than in wet years in Bolinas Lagoon [38] and Tomales Bay [37]. It is also possible that the large numbers of least sandpipers wintering in San Francisco Bay may have masked movements from the Central Valley, where their mean annual count was an order of magnitude lower in our study. Combined, the responses of dunlins and least sandpipers suggest potential resilience to drought on broad scales, but the long-term effects, if any, on population size remain unclear. Better information is needed on the costs incurred when shorebirds shift distributions during the non-breeding season to understand the potential impacts on their populations.

We found consistently lower abundance of dowitchers during the period of 3-year drought across interior habitats and coastal estuaries of central California, which comprise an important wintering area for non-breeding long-billed dowitchers [31, 60]. Dowitchers tend to gather in flocks and probe in mud when foraging similar to *Calidirs* sandpipers, but we found evidence of declines during drought across our study area rather than shifting patterns of abundance. Movement distances of long-billed dowitchers tracked habitat availability at broader scales in the Central Valley [29], which suggests some dowitchers may have moved beyond our study area in response to drought. On the other hand, past studies of radio-tagged long-billed dowitchers found high fidelity during winter to San Francisco Bay [31] and moderate fidelity to the Central Valley [29], which suggests that regional population declines during drought may be indicative of broader declines in dowitcher populations. Additional assessments of population trends are needed for both dowitcher species, as well as the dunlin, because they are considered highly vulnerable to climate change [61], and of moderate (dunlin and long-billed dowitcher) to high (short-billed dowitcher) conservation concern [62].

We present robust findings related to the effects of drought on shorebirds despite limitations of our sampling design and logistical constraints of surveying for shorebirds over an extended time periods at a broad scale. For example, our estimates of abundance of killdeers, more than other focal species, may have been influenced by movements within regions or sub-

regions to miscellaneous habitats that we did not survey extensively, given their relatively high use of miscellaneous land cover types. Surveying over a broad scale resulted in variability in the total annual area surveyed because inevitably each year access at some survey units was limited by road conditions or other logistical constraints. However, annual variation in survey effort was not related to our classification of drought years and thus we considered any impact on our results to be negligible. Lastly, obtaining accurate counts of large aggregations of shorebirds is challenging even for highly experienced surveyors. Thus, we suspect that counts of larger aggregations of shorebirds were less accurate than smaller aggregations, which may have influenced results in San Francisco Bay more than in the Central Valley or Point Reyes regions; aggregations are especially large around high tide in San Francisco Bay when most intertidal mudflats are covered by sea water.

## Conclusions

In our study nonbreeding shorebirds responded to drought by either utilizing the remaining flooded habitats in the Central Valley or by redistributing among a network of interior and coastal habitats. The presence of legally mandated but reduced deliveries of fresh water to refuge wetlands [34], government funded incentive programs to flood wetlands and agricultural lands, and targeted shallow flooding for shorebirds [63] did not preclude a substantial reduction in populations of nonbreeding shorebirds during drought in the Central Valley. If we assume that the total population of shorebirds in the Central Valley is still equivalent to results reported by [35], based on our models that estimated a reduction in shorebird abundance of approximately 25% during drought years, 50,000 to 75,000 shorebirds may have departed the Central Valley. Yet our findings also suggest that the combined impacts of governance and conservation efforts may have delayed or mitigated the effects of drought in 2013 and 2014. During future droughts, additional deliveries of fresh water for wildlife will be necessary to better compensate for reduced flooding of wetland habitats on the landscape. Future conservation targets for flooded habitat should consider the amount of flooding within shorebird habitats in the Central Valley during December 2014 [24] to be a minimum threshold below which there may be widespread declines in shorebird abundance. Shorebirds and other wetland-dependent wildlife will also benefit from increased use of the shallow-water flooding practices in agricultural lands [64] and managed wetlands [65].

Conservation and management efforts can further reduce the effects of drought on shorebirds by creating, restoring, or expanding flooded habitats as drought refugia. For example, we observed shorebirds congregating in the Grasslands Ecological Area of the San Joaquin sub-region during 2013, and then at three locations near the Sacramento River in 2014, including the Yolo bypass area of the Yolo-Delta sub-region. The Tulare sub-region had the greatest reductions in freshwater deliveries for flooding wetlands of any sub-region in the Central Valley, and our study confirmed that refuges in Tulare did not function as drought refugia for shorebirds. Additional fresh water for wildlife in Tulare during future droughts may be imperative to reach and maintain population targets for shorebirds in the Central Valley. During future droughts, coastal habitats may play an even more important role in maintaining a healthy network of wetlands across central California if extreme drought years occur more often. On the coast intertidal mudflats are drought tolerant, and recent broad-scale restoration and management of coastal wetlands has shown potential to increase carrying capacity for shorebirds within San Francisco Bay [66]. Additional research is needed to better understand the effects of drought on long-term trends of shorebird populations among regions of central California and how regional effects of drought may be related to broad-scale trends in shorebird populations.

Lastly, our results show that shorebirds use a highly connected network of interior and coastal wetlands across central California in many different ways. A multi-scale monitoring framework was necessary to robustly investigate the effects of drought on non-breeding shorebirds [6]. Similarly, conservation planning efforts must occur at several spatial scales to be effective for shorebird populations as a whole. While it is likely not feasible to design any individual conservation action to benefit all shorebirds, maximizing the net benefit of a conservation action will require coordination with other similar actions across a broad landscape. During future droughts, a strategic and coordinated response should involve government and conservation organizations acting at the local, regional, and continental levels. Moreover, expanding and strengthening communication networks for conservation across multiple levels of governance further promotes a healthy functioning network of wetlands across the Pacific Coast of the Americas.

## Supporting information

**S1 Fig. The annual mean proportion flooded of shorebird survey units by land cover type.** Annual means with error bars representing 95% confidence intervals from data collected during shorebird surveys from 2011 to 2016 in the Central Valley, California, USA. Survey units are categorized by the dominant land cover type within unit boundaries. The dashed line represents the annual mean estimate across all cover types. See Methods for details on the types of land cover that composed the "other suitable" and "miscellaneous" land cover categories.
(TIF)

**S2 Fig. The annual mean proportion flooded of shorebird survey units by sub-region in the Central Valley.** Annual means with error bars representing 95% confidence intervals from data collected during shorebird surveys from 2011 to 2016 in the Central Valley, California, USA. Survey units are categorized by geographic location within sub-regions. The dashed line represents the annual mean estimate across all cover types.
(TIF)

**S1 Table. Annual density of total shorebirds (per km2) and survey effort across central California, USA.** Estimated density of total shorebirds (per $km^2$) among four sub-regions of the Central Valley and two coastal regions from 2011 to 2016 across central California, USA. We did not estimate the mean density of total shorebirds within coastal regions. Summary of our annual survey effort is located under each density estimate and displayed as the number of units and total area surveyed ($km^2$). [1] No survey at Los Banos Wildlife Area in 2014. [2] No survey at Kern or Pixley National Wildlife Refuge in 2011 and 2014. [3] No survey of Suisun Marsh in 2011. [4] No surveys of Drakes Estero or nearby waters in 2011 and 2012.
(DOCX)

**S2 Table. Results of models of annual abundance of focal shorebirds in central California, USA.** Results of generalized linear mixed models of abundance of six focal shorebirds among three regions of central California, USA. Data collected during annual shorebird surveys from 2011 to 2016. Models tested differences in abundance between non-drought years and drought years (2013 to 2015 for drought models; 2014 to 2015 for extreme drought models). Parameter estimates (β) and 95% confidence intervals (CI) are back-transformed to the scale of the response variable. *P*-values are from a null-hypothesis likelihood-ratio test on the t-statistic for each parameter. Statistically significant *p*-values are in italics. Deviance scores estimate fit to the data for each model; a lower score represents better fit.
(DOCX)

**S3 Table. Annual density of total shorebirds (per km2) and survey effort in the Central Valley.** Estimated density of total shorebirds (per $km^2$) among four land cover types from 2011 to 2016 in the Central Valley, California, USA. Summary of our annual survey effort is located under each density estimate and displayed as the number of units and total area surveyed ($km^2$). See Methods for details on the types of land cover that composed the "other suitable" and "miscellaneous" land cover categories.
(DOCX)

**S4 Table. Annual density (per km2) of focal shorebirds and mean annual count in the Central Valley.** Annual density estimates among four dominant land cover types from 2011 to 2016 in the Central Valley, California, USA. Mean annual counts are the average annual survey totals for each species without accounting for the total area surveyed.
(DOCX)

**S5 Table. Annual density (per km2) of non-focal shorebirds and mean annual count in the Central Valley.** Annual density estimates from 2011 to 2016 in the Central Valley, California, USA. Mean annual counts are the average annual survey totals for each species without accounting for the total area surveyed.
(DOCX)

**S6 Table. Annual density (per km2) of focal shorebirds and mean total count in central California, USA.** Annual density estimates from 2011 to 2016 among three regions in central California, USA. Mean annual counts are the average annual survey totals for each species without accounting for the total area surveyed.
(DOCX)

**S7 Table. Results of models of the proportion flooded of survey units in central California, USA.** Results of generalized linear mixed models of the proportion flooded of survey units among four land cover types in the Central Valley and three regions of central California, USA. Data collected during annual shorebird surveys from 2011 to 2016. Models tested differences in abundance between non-drought years and drought years (2013 to 2015 for drought models; 2014 to 2015 for extreme drought models). Parameter estimates (β) and 95% confidence intervals (CI) are back-transformed to the scale of the response variable. *p*-values are from a null-hypothesis likelihood-ratio test on the t-statistic for each parameter. Statistically significant *p*-values are in italics. Deviance scores estimate fit to the data for each model; a lower score represents better fit.
(DOCX)

**S1 Dataset. Complete dataset from surveys of shorebirds and their habitats from 2011 to 2016 in central California, USA.** Dataset used for summaries and analyses of shorebird abundance and habitat availability in central California. Each line in the dataset represents an individual survey at a survey unit. Columns are composed of factors related to the survey unit and results from the survey, including counts of shorebirds and an estimated proportion of the survey unit that was flooded.
(CSV)

## Acknowledgments

This study was made possible by contributions from a network of government agencies, conservation foundations and organizations, and volunteers. Government agency contributions were led by Matthew Fries, Geoffrey Grisdale, Cheryl Strong, and Mike Wolder of the U.S.

Fish and Wildlife Service, David Press and Matthew Lau of the National Park Service, Tanya Graham and Stacy Moskal of the U.S. Geological Survey, Laura Cockrell, Orlando Rocha, Lara Sparks, and Dave VanBaren of California Department of Fish and Wildlife, and Chris Conard of the Sacramento Regional Sanitation District Water Resources. We also thank Ben Pearl of San Francisco Bay Bird Observatory, Nils Warnock and Emiko Condeso of Audubon Canyon Ranch, Ric Ortega of Grassland Water District, Sara Sweet of The Nature Conservancy, Samantha Arthur, Meghan Hertel and Andrea Jones of Audubon California, and Kristin Sesser, Mark Dettling, and Gary Page of Point Blue Conservation Science for their contributions. Most importantly we thank the many other individuals not listed here, primarily volunteers whose tireless contributions made this study possible. This is Point Blue contribution number 2333.

## Author Contributions

**Conceptualization:** Blake A. Barbaree, Matthew E. Reiter, Khara M. Strum.

**Data curation:** Blake A. Barbaree, Matthew E. Reiter, Jennifer E. Isola, L. Max Tarjan, Lynne E. Stenzel, W. David Shuford.

**Formal analysis:** Blake A. Barbaree, Matthew E. Reiter.

**Funding acquisition:** Matthew E. Reiter, Catherine M. Hickey.

**Investigation:** Blake A. Barbaree, Matthew E. Reiter.

**Methodology:** Blake A. Barbaree, Matthew E. Reiter, Khara M. Strum.

**Project administration:** Blake A. Barbaree, Matthew E. Reiter, Khara M. Strum, W. David Shuford.

**Resources:** Blake A. Barbaree, Matthew E. Reiter, Catherine M. Hickey, Jennifer E. Isola, Scott Jennings, L. Max Tarjan, Cheryl M. Strong, Lynne E. Stenzel, W. David Shuford.

**Supervision:** Blake A. Barbaree, Catherine M. Hickey.

**Validation:** Blake A. Barbaree.

**Visualization:** Blake A. Barbaree.

**Writing – original draft:** Blake A. Barbaree.

**Writing – review & editing:** Blake A. Barbaree, Matthew E. Reiter, Catherine M. Hickey, Khara M. Strum, Jennifer E. Isola, Scott Jennings, L. Max Tarjan, Cheryl M. Strong, Lynne E. Stenzel, W. David Shuford.

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
