## [Decision Letter · Decision Letter 0]

14 Aug 2020

PONE-D-20-20377

Effects of drought on the abundance and areas of concentrations of non-breeding shorebirds in central California,

PLOS ONE

Dear Dr. Barbaree,

Thank you for submitting your manuscript to PLOS ONE. After careful consideration, we feel that it has merit but does not fully meet PLOS ONE’s publication criteria as it currently stands. Therefore, we invite you to submit a revised version of the manuscript that addresses the points raised during the review process.

We received thorough and thoughtful reviews from two reviewers with expertise in shorebird ecology. Your paper is generally well-written; however, as you will see in comments from both reviewers, the message you are trying to convey to the reader is not easy to follow. The Discussion is quite long and repeats a lot of the results rather than discussing the findings. The paper could be improved by taking a step back to determine a clear message and then re-structuring the content appropriately. Both reviewers provide helpful feedback on re-framing and messaging both for the Discussion and for the paper as a whole. In addition to the review comments below, you will find three attachments with more detailed comments.

We look forward to receiving your revised manuscript.

Kind regards,

Stephanie S. Romanach, Ph.D.

Academic Editor

PLOS ONE

Journal Requirements:

2)  We note that you have indicated that data from this study are available upon request. PLOS only allows data to be available upon request if there are legal or ethical restrictions on sharing data publicly. For more information on unacceptable data access restrictions, please see http://journals.plos.org/plosone/s/data-availability#loc-unacceptable-data-access-restrictions.

Reviewers' comments:

Reviewer's Responses to Questions

**Comments to the Author**

1. Is the manuscript technically sound, and do the data support the conclusions?

Reviewer #1: Yes

Reviewer #2: Yes

2. Has the statistical analysis been performed appropriately and rigorously? 

Reviewer #1: Yes

Reviewer #2: Yes

3. Have the authors made all data underlying the findings in their manuscript fully available?

Reviewer #1: No

Reviewer #2: Yes

4. Is the manuscript presented in an intelligible fashion and written in standard English?

Reviewer #1: Yes

Reviewer #2: Yes

5. Review Comments to the Author

Reviewer #1: This was a well written paper which explores the changes in abundance, density and distribution of shorebirds in the central valley of California during drought.

The literature has rarely looked at impacts of drought on shorebirds, and there are only a couple examples of evidence of shorebirds shifting from inland wetlands to the coast when conditions become unsuitable. For both these reasons this paper presents some useful results, quantifying changes in abundance from wetlands that are likely to become increasingly threatened.

Overall, I think this manuscript would benefit from choosing a central message and focusing more on it. It may help to focus, for example, on the number of birds, and amount of habitat estimated to have been lost during the drought. It is currently not easy to form those kinds of overall estimates as a reader. What is the magnitude of the changes. Regardless of what might be selected to be the main message / result of the paper. Results and discussion might be strengthened by focusing on a main message in the first couple of paragraphs of these sections. As currently written, these sections seem to wander a bit. The connection of shifts in distribution and abundance to potential impacts to shorebirds are not yet as clear as they could be in these locations.

There are a couple of points raised which do not have strong supporting evidence within the study, and it would be good to focus less strongly on those.

On a technical note, it would be good to indicate that there were no influential outliers in the methods. In recent work on shorebird data we found NB and ZINB models did not capture changes in total abundance when there were large outliers in the data. In our case while NB models captured differences in means well, they masked the big changes in abundance from year to year reflected in the outliers. I suspect this was not a problem here, but just wanted to raise the possibility as it caught me by surprise given some nice residual plots. Summing the total annual counts would be a good way to check that the patterns are the same whether using NB means or totaled counts.

I would strongly suggest presenting the statistical equation representing models used with terms defined.

It would help explain parameter estimates, high deviance values and confirm the terms used.

In the attached PDF are some minor comments and suggestions.

Reviewer #2: Thanks for the opportunity to review this manuscript. Barbaree et al. present a summary of an impressive survey effort, a long-term assessment of nonbreeding shorebird numbers in the Central Valley and nearby coastal estuaries of California. These data represent the effort and coordination of numerous participants, and constitute a valuable body of work to help understand factors affecting the abundance and distribution of nobreeding shorebirds. The authors specifically couch their results in the context of California’s recent droughts, and explore how variation in surface water across various habitats influences the occurrence of shorebirds. I applaud the authors for their dedication and perseverance in collecting and analyzing this impressive dataset.

In general, I found the manuscript to be well written and easy to read, and most of my comments are editorial in nature. I do feel that the authors could strengthen the manuscript in many ways, however, efforts that I outline below. Specifically, I found the discussion to be long and rather repetitive, mostly a summary of the results, when I think an exploration of causal patterns would be more enlightening (and interesting). To this end, I believe that the authors first need to better make their case for shifting numbers between (and among) Central Valley and coastal sites being due to displacement and resettlement patterns. While this seems obvious, the authors omit the large body of work (specifically on Dunlin, Killdeer, and dowitchers—see suggested references in attached comments) tracking the within-season movements of shorebirds in these regions. The authors cite [23] and [24] in the Introduction, but the documented movements of these species deserve more elaboration, either in the Introduction or Discussion. A few sentences summarizing some of this work would help the reader better appreciate the ways in which these birds likely are resettling in response to landscape-scale factors. This is important, because the authors use terms like ‘vulnerability’ throughout the paper, suggesting that perhaps certain species are suffering detrimental effects due to the drought conditions. The authors provide no evidence that year-to-year fluctuations in numbers reflect population trends, or that these changes in numbers reflect negative impacts to the species; indeed, such response could instead be viewed as resilience instead of vulnerability. I would suggest a less loaded term to replace vulnerability—‘sensitivity’? Second, I think the authors need to spend some time convincing the reader why their two time periods (drought: 2013-2015 & extreme drought: 2014-2015) are relevant. As figures 3-5 indicate, the differences between these two factors are pretty inconsequential, with large overlap in confidence intervals, and nearly equal point estimates, between these periods. There are some differences in species-specific responses (Table 1), but in general the trends are similar, and there’s no real information on why examining the results as such (drought & extreme drought) is meaningful biologically. I think the story would be simplified were the authors to simply present one or the other, rather than both. This would simplify the story. As it stands, there is no convincing explanation of how these two periods are meaningfully different. This ultimately leads to the third topic that I believe would help improve the discussion. There is little exploration of the functional response of the various species. Why would yellowleg densities in rice fields decrease in extreme drought years but not drought years? If such distinctions aren’t explored, the distinctions are seemingly arbitrary to the reader. So, I’d suggest simplifying the story by just retaining one period, and then use the discussion to explore the functional traits of the species in question that may reflect the variation (or lack thereof) in their numbers. Is it water depth? Soil moisture? Feeding guild? The feeding ecologies of Dunlin and Killdeer are very different, for instance, and the discussion would be much more interesting to the reader were the authors to explore some of these ecological traits, even speculatively, to help explain the patterns in abundance and distribution that they detected.

I hope you find these comments useful, and look forward to seeing these results in the scientific literature soon. Please feel free to contact me if you have questions or comments on my suggestions. Sincerely,

Dan Ruthrauff

USGS Alaska Science Center

druthrauff@usgs.gov

6. PLOS authors have the option to publish the peer review history of their article (what does this mean?). If published, this will include your full peer review and any attached files.

Reviewer #1: No

Reviewer #2: **Yes: **Dan Ruthrauff

---

## [Author Response · Author response to Decision Letter 0]

28 Sep 2020

Reviewer #1: This was a well written paper which explores the changes in abundance, density and distribution of shorebirds in the central valley of California during drought.

• Response: The authors send thanks to the reviewer for their thoughtful review and different perspective on several accounts. Below we have addressed each comment separately.

The literature has rarely looked at impacts of drought on shorebirds, and there are only a couple examples of evidence of shorebirds shifting from inland wetlands to the coast when conditions become unsuitable. For both these reasons this paper presents some useful results, quantifying changes in abundance from wetlands that are likely to become increasingly threatened.

Overall, I think this manuscript would benefit from choosing a central message and focusing more on it. It may help to focus, for example, on the number of birds, and amount of habitat estimated to have been lost during the drought. It is currently not easy to form those kinds of overall estimates as a reader. What is the magnitude of the changes. Regardless of what might be selected to be the main message / result of the paper. Results and discussion might be strengthened by focusing on a main message in the first couple of paragraphs of these sections. As currently written, these sections seem to wander a bit. The connection of shifts in distribution and abundance to potential impacts to shorebirds are not yet as clear as they could be in these locations.

• Response: We thank the reviewer for these very helpful suggestions. They were used to guide revisions to the results and discussion in particular. We focused on a main theme of all shorebirds declining in the interior region. We included percentage estimates for changes in abundance and flooding throughout the revisions to provide context for magnitude of the changes. Although we refrained from presenting estimations of total shorebird numbers because ours was not a comprehensive survey, we did include a rough estimate of total shorebirds that departed the Central Valley in the Conclusions section to provide the context that may have been missing previously.

There are a couple of points raised which do not have strong supporting evidence within the study, and it would be good to focus less strongly on those.

• Response: Agree and we de-emphasized several points that were flagged by both reviewers.

On a technical note, it would be good to indicate that there were no influential outliers in the methods. In recent work on shorebird data we found NB and ZINB models did not capture changes in total abundance when there were large outliers in the data. In our case while NB models captured differences in means well, they masked the big changes in abundance from year to year reflected in the outliers. I suspect this was not a problem here, but just wanted to raise the possibility as it caught me by surprise given some nice residual plots. Summing the total annual counts would be a good way to check that the patterns are the same whether using NB means or totaled counts.

• Response: We agree that influential outliers are important to consider when modelling shorebird count data. Our distribution plots were okay, and we qualitatively determined patterns in the count data sufficiently matched model results. We considered each of the top 20 outliers for total shorebirds in the Central Valley, and we highlight three outlier counts in the results and discussion that occurred during 2014. Models on dunlin found no differences in abundance in drought vs nondrought years and if the 2014 outliers were inappropriately influential then we might have suspected a significant increase of abundance in the Central Valley. In addition, we conducted a quantitative comparison of model fit to determine negative binomial was more appropriate than poisson and zero-inflated models, but this assessment was not included in the manuscript for brevity and because we considered the subject repetitive with previous studies. In the end, we found some tradeoffs with different models types, but negative binomial models had clearly the beast performance all things considered.

I would strongly suggest presenting the statistical equation representing models used with terms defined.

• Response: We included the model structure as a formula in the revision.

It would help explain parameter estimates, high deviance values and confirm the terms used.

• Response: We expanded the last paragraph of the methods to include more details on the models including the model formula and explanations for model parameters and deviance values. Explanations of parameters were included in the original manuscript, but references to deviance had been confined to the table captions.

In the attached PDF are some minor comments and suggestions.

Reviewer #1: minor comments

Title: "on the abundance, density and distributions"? I suggest changing this phrase throughout. Just akward.

• Response: We agree the phrase “areas of concentration” can be awkward. We have replaced “areas of concentration” with “distribution” throughout the revision to increase clarity for readers, and we also included a further context that we were assessing broad-scale distributions and the regional and sub-re4gional scales. Additionally, we did not include “density” in the title because within the paper we explain that density was used as a proxy for abundance in the study.

L8: I think it would really strengthen this ms if the magnitude of this change could be expressed. % or better total individuals estimated to have disappeared from the central valley (I'm guessing it would be more than a 100 birds, but it is not easy to do more than guess at the magnitude from the figures reported)

• Response: We better highlighted the fact that β results from models represent the proportional change in abundance from non-drought to drought years. We now use % change estimates throughout the revised paper and provide a rough estimate of the number of shorebirds that departed the Central Valley by applying our findings to Shuford et al. (1998) that estimated shorebirds population nearly 30 years ago.

L9-10: Perhaps delete or put in context of the main message

• Response: We removed this statement from the revised abstract.

L14-15: I would not mention this in the abstract, and just raise the possibility in the discussion. I did not think there was much evidence that drought was the cause of the drop, just one of the possible explanations. Alternatively strengthen the evidence of this possibility in the ms.

• Response: We removed this statement from the revised abstract.

L16-19: Perhaps change to something like: Changes in abundance suggest some (magnitude if it can be reported) dunlin and least sandpiper shifted to the coast in response to drought.

• Response: We revised this statement to include magnitude of the changes in abundance for both species.

L19-20: Suggest deleting this sentence unless evidence could be strengthened. I think estimating the magnitude of these changes, how many birds, how much habitat might be good to add as a sentence here. Alternatively, a sentence that focuses on the most important result (some options in the conclusion), or potential implications that your work point toward. When I read this I was struck at how little the drought seemed to change habitat availability and shorebird numbers (less than I expected, but I may be missing the true magnitude of the changes). It was also great to see some evidence of movement of shorebirds from inland to coastal wetlands. This has not been widely reported in the literature.

• Response: We replaced this sentence with potential conservation implications, in part referring to sentiment noted in the comment below on L54.

L54: I think most readers would expect changes in abundance and distribution at a local scale as a result of drought. It is worth highlighting that such changes could point toward population impacts. It is also worth highlighting how changes to abundance might impact capacity to hit the conservation targets mentioned in the conclusion.

• Response: We now reference conservation targets in this portion of the introduction and thank the reviewer for this suggestion.

L138-141: Curious if the error associated with the method has been quantified in Page or elsewhere (other readers might be interested if it is available).

• Response: This is a great question, and the short answer is no. In lieu of an estimate of error, we included the proportion of total Calidris sandpipers counted that occurred within unidentified mixed flocks in each region, which was similar to the context provided in Page et al.

L158-163: In our region, duck lakes can be suitable in a drought, and temporary wetlands can support large concentrations of shorebirds when they become suitable (something more likely to happen outside the drought). If it is a possible issue, it might be worth a sentence regarding the assumptions in this study.

• Response: We included an additional sentence to address this concept in the paragraph in the discussion that focuses on limitations of the study. The topic sentence for that paragraph is as follows “We also acknowledge limitations of our sampling design and logistical constraints of surveying for shorebirds over an extended time periods at a broad scale.”

L272-273: Statistical notation of the model used with terms defined would help inform readers what was done.

• Response: We included the suggested notation in the final paragraph of the methods in the revised manuscript.

L284: This is good, but I would lead with the most important results or the ones you want to focus on most. Ordering results from most to least important may help focus the results.

• Response: We rearranged the results in the revision to lead with our models on total shorebirds, followed by models on focal species and habitat availability.

L319: This could be the main and strongest result, and might move it to the start. Again, the magnitude of this results in estimated total number of shorebirds in the central valley would likely help show the importance of this result. Even an estimate of percent decline in drought, and then simply applying that percent to the best available estimate of the shorebird population in the central valley would be informative.

• Response: See previous response about restructuring the results section. As for reporting magnitude, we now report the magnitude of change from non-drought to drought years thought using percentage change and a rough estimate of birds that departed the Central Valley during drought.

L390: Suggest leading this paragraph with this result, it seems the most important but is a bit buried under non-significant results.

• Response: We restructured this paragraph to lead with statistically significant results.

L400: Again, what does this translate to in terms of ha of wetland habitat.

• Response: We included the estimated total amount of flooded habitat in the revision as well as percent declines in flooding within each land cover type.

L430: It does not seem there is alot of agreement between results in Wen, and these. They seem much weaker here, that is clarified in the next sentence, but even the drop in habitat suitablility seems comparatively modest. Suggest rephrase.

• Response: We adjusted the interpretation of these studies in the revisions. WE now point out the difference in magnitude of changes between studies and elucidate potential causal factors including differences in climate and sampling design of the studies.

L433: This was not obvious to me from the results. Perhaps clarify the line of evidence here in a sentence or two.

• Response: Both reviewers flagged this comment. We removed the use of the term vulnerability and instead categorized these findings as showing population declines within our study area only.

L441: Perhaps better to say "Not unlike", as this does not appear similar to Australia to me.

• Response: We revised this entire statement

L458: Australia often sees a two year lag from filling to peak shorebird abundance. It often takes time for invertebrates to establish and then get pushed into higher densities through wetland drawdown.

There are better references of this, but here is one: Chambers, L.E. & Loyn, R.H. (2006) The influence of climate variability on numbers of three waterbird species in Western Port, Victoria, 1973-2002. International Journal of Biometeorology, 50, 292-304.

Bino or Kingsford have used the lag at some point as well.

• Response: We thank the reviewer for sharing this insight and we included this reference for a lagged response in the revision.

L465: Yes agree, the more dynamic things become the less local scale seems to reflect the big picture.

• Response: We made this message clearer in the revision.

L506: It would be great to have an estimate of the total number of shorebirds thought to have gone missing in the central valley as a result of the drought. Are we talking 1,000 birds, or 10,000 or 100,000?

• Response: See previous comments regarding our inclusion of magnitude of change and an estimate of total birds departing the Central Valley.

L506: It would also be informative to have a rough estimate of how much habitat decreased (10,000 ha)?

• Response: We included estimates of area of flooding in the results section. This portion of the discussion was removed from the revision, but the comment helped us provide more information within the results.

L513: and perhaps that optimal management needs to consider interspecific differences in both habitat and movement patterns.

• Response: We agree and included this idea in the conservation implications given the amount of interspecies variation observed during our study.

L527: A rough estimate of the number of individuals that left inland areas, and the number that showed up in the coast would really strengthen this possibility.

• Response: We agree this information has potential to strengthen our findings, so we included references to our mean counts of dunlin to show that increases in the San Francisco Bay area were large enough to suggest that dunlin moved there from many areas including beyond our study area. That said, we generally refrain from referring to counts of shorebirds because ours was not a comprehensive survey and thus estimates of birds counted could be misleading.

L535: In Australia there are examples of wetlands that only become exceedingly good for shorebirds during drought, as evaporation and wetland drawdown increases availability and denisty of invertebrates. Duck lakes can hold tens of thousands of shorebirds when the drought lasts long enough.

• Response: This is an important idea when considering redistribution of shorebirds during drought. We agree that our sampling frame had potential to exclude a few locations that may become more important during drought, but only a comprehensive survey could cover all potential areas of importance. We remain confident that our focus on areas with a high likelihood of flooding captured most locations likely to be important for shorebirds during a drought. Examples of areas of potential importance that may have been excluded from our sampling frame included smaller wetland complexes of private lands that were spatially isolated from similar habitats, and some wastewater treatment plants and reservoirs. That said, we feel that our sampling design was sufficient to document abundance at broad scales despite not being entirely comprehensive.

 

Reviewer #2: Thanks for the opportunity to review this manuscript. Barbaree et al. present a summary of an impressive survey effort, a long-term assessment of nonbreeding shorebird numbers in the Central Valley and nearby coastal estuaries of California. These data represent the effort and coordination of numerous participants, and constitute a valuable body of work to help understand factors affecting the abundance and distribution of nonbreeding shorebirds. The authors specifically couch their results in the context of California’s recent droughts, and explore how variation in surface water across various habitats influences the occurrence of shorebirds. I applaud the authors for their dedication and perseverance in collecting and analyzing this impressive dataset.

In general, I found the manuscript to be well written and easy to read, and most of my comments are editorial in nature. I do feel that the authors could strengthen the manuscript in many ways, however, efforts that I outline below. Specifically, I found the discussion to be long and rather repetitive, mostly a summary of the results, when I think an exploration of causal patterns would be more enlightening (and interesting). To this end, I believe that the authors first need to better make their case for shifting numbers between (and among) Central Valley and coastal sites being due to displacement and resettlement patterns. While this seems obvious, the authors omit the large body of work (specifically on Dunlin, Killdeer, and dowitchers—see suggested references in attached comments) tracking the within-season movements of shorebirds in these regions. The authors cite [23] and [24] in the Introduction, but the documented movements of these species deserve more elaboration, either in the Introduction or Discussion. A few sentences summarizing some of this work would help the reader better appreciate the ways in which these birds likely are resettling in response to landscape-scale factors. This is important, because the authors use terms like ‘vulnerability’ throughout the paper, suggesting that perhaps certain species are suffering detrimental effects due to the drought conditions. The authors provide no evidence that year-to-year fluctuations in numbers reflect population trends, or that these changes in numbers reflect negative impacts to the species; indeed, such response could instead be viewed as resilience instead of vulnerability. I would suggest a less loaded term to replace vulnerability—‘sensitivity’? Second, I think the authors need to spend some time convincing the reader why their two time periods (drought: 2013-2015 & extreme drought: 2014-2015) are relevant. As figures 3-5 indicate, the differences between these two factors are pretty inconsequential, with large overlap in confidence intervals, and nearly equal point estimates, between these periods. There are some differences in species-specific responses (Table 1), but in general the trends are similar, and there’s no real information on why examining the results as such (drought & extreme drought) is meaningful biologically. I think the story would be simplified were the authors to simply present one or the other, rather than both. This would simplify the story. As it stands, there is no convincing explanation of how these two periods are meaningfully different. This ultimately leads to the third topic that I believe would help improve the discussion. There is little exploration of the functional response of the various species. Why would yellowleg densities in rice fields decrease in extreme drought years but not drought years? If such distinctions aren’t explored, the distinctions are seemingly arbitrary to the reader. So, I’d suggest simplifying the story by just retaining one period, and then use the discussion to explore the functional traits of the species in question that may reflect the variation (or lack thereof) in their numbers. Is it water depth? Soil moisture? Feeding guild? The feeding ecologies of Dunlin and Killdeer are very different, for instance, and the discussion would be much more interesting to the reader were the authors to explore some of these ecological traits, even speculatively, to help explain the patterns in abundance and distribution that they detected.

I hope you find these comments useful, and look forward to seeing these results in the scientific literature soon. Please feel free to contact me if you have questions or comments on my suggestions. Sincerely,

Dan Ruthrauff

USGS Alaska Science Center

druthrauff@usgs.gov

• Response: We thank this reviewer for their thorough evaluation and experienced insight on ways to improve the paper. We have addressed all of the suggestions above and those in the minor comments. Below we have separate responses to the issues raised in reviewer 2’s summary comments above.

o We agree that the discussion needed to be improved and we made substantial revisions based on this comment and those from reviewer #1. Our primary goal was to improve cohesiveness and topical focus for the reader. In doing so we removed most of the statements that reviewed results and increased focus on causal factors mostly related to species ecology. We also revised the introduction to better highlight past research that documented movements between our study regions, and we included the suggested references from reviewer 2 to provide additional support for references already included in the study. To that end, we revised or removed conclusions that used loaded terms like vulnerability because our study was not able to assess population status beyond central California.

o We agree that improving our justification for using two separate drought periods would improve the manuscript, and focusing on only one drought period would simplify the story. Generally, we prefer a simplified story and focusing on only the 3-year drought period would not change the take home messages. That said, we felt it was important to investigate multiple temporal classifications of drought because our study included multiple species with different ecological tendencies, and curtailments of freshwater for wildlife peaked in 2014-2015. The primary example that resulted in us deciding to retain both drought periods in the revision was that removing the 2-year would have excluded all findings related to effects of drought on yellowlegs, when in fact our data suggested otherwise at both interior and coastal sites. In addition to ecological differences among species, we are studying a system in which water policy can drive habitat availability as much if not more than the actual amount of precipitation. Thus multiple years of drought can result in quick large changes to habitat not simply because of lack of precipitation but legal decisions to further reduce the water that exists. For example in 2014 and 2015 term 91 was enacted which limited the water available to flood rice post-harvest. 

Reviewer #2: minor comments to text

L9-10: observed? recorded? 'found evidence of' makes it sound like you detected footprints in the mud or something...

• Response: We changed the wording here and in other areas where we were not referring to statistical evidence.

L69: I prefer the plural (stilts v. stilt) in this situation...but at least be consistent: as 'yellowlegs' is always plural, sounds better to make others plural, too. And I'm a believer in the Warnockian 'dunlins', 'killdeers' (and whimbrels, for that matter!) for species for which the singular is often used in plural. :-)

I'll not edit these situations hereafter, but note and maintain consistency throughout

• Response: Ultimately the authors do not feel strongly either way, so we revised our use of species names to be plural in all occasions where we were not speaking directly about species.

L226: Now, here I'd actually keep these as singular, as you preceded all with 'the'...

• Response: See response above. We revised most instances where “the” was used to proceed a species name, but it was retained in a few occasions where we were speaking directly about species, rather than abundance, distribution, or other results.

L295: As a note, the table would be easier to read were the land cover types (ie, wetlands, Rice fields, etc) centered at the top of their respective cells rather than the middle...hard to see where one group of species ends and another starts as currently formatted.

• Response: We revised all tables as suggested.

L321: Is this meant to read as '...of 3-year drought (β = 0.78, 95% CI: 0.63, 0.98, p = 0.03) and 2-year extreme drought (β = 0.71, 95% CI: 0.56, 0.91, p = 0.007; Fig 3) than in non-drought years'? ie, should the second parenthetical be moved immediately behind '2-year extreme drought', as the figure would seem to suggest? As currently ordered, the Beta of .71 confusingly seems to pertain to non-drought years (which is the reference level)

• Response: We included this revision as suggested.

L329-330: As above, should not the second parenthetical immediately follow '2-year extreme drought' rather than 'non-drought years' to avoid confusion?

• Response: We included this revision as suggested.

L427-429: This example comes at the reader as a bit random: southeastern Australia? If you find other similar results from other regions, you could broaden your statement, but to cite this one study so specifically without further explanation comes a bit out of left field.

• Response: We moved this to the end of the paragraph and provided more context. We reference Australia twice in the discussion because this region has more published research on waterbirds and drought than any other (to our knowledge).

L433: Hmm, perhaps consider 'sensitivity' instead of vulnerability. You don't know what happened to the yellowlegs or dowitchers, only that there were fewer across both regions during the drought years. Perhaps they happily migrated elsewhere, and aren't vulnerable at all; this could instead be interpreted as resiliency, not vulnerability. So, a word like 'sensitivity', without negative connotations, is probably more appropriate.

• Response: We revised this statement to simply suggest that some shorebirds do not respond to drought by shifting to nearby coast regions, rather than implying population level consequences.

L433: 'drought effects' is pretty vague: you mean changes in abundance, no? I would be as specific as possible.

• Response: We revised wording in several locations of the text in response to this comment.

L441: Again, oddly specific. Surely there must be other examples of variation in abundance of birds as a function of wetland cover?

• Response: Literature on shorebirds and drought is scarce, outside of Australia and limited work in North America that has occurred mostly during migration.

L471, L484: ‘drought effects’

• Response: See response to similar comment on L433

L482: Again, consider another term: sensitivity or similar

• Response: We removed the term “vulnerability” from the manuscript entirely.

L568-569: Hmm. Galbratih et al.'s projections are with respect to climate change; 'highly imperiled' seems alarmist. Seems like you should state something regarding their current conservation status. The latest species rankings from the US Shorebird Conservation Partnership (https://www.shorebirdplan.org/wp-content/uploads/2016/08/Shorebirds-Conservation-Concern-2016.pdf) list pacifica Dunlin and LBDO as 'Moderate Concern'; SBDO are listed 'High Concern'.

• Response: We thank the reviewer for pointing out potentially misleading detail. In the revision we now refer to Galbraith’s projections as related to climate change, and then refer to the species status as listed by USSCP.

L610: How? While I agree that long-term conservation efforts help shorebirds, you've not demonstrated how these efforts are reflected in your results. What about long-term conservation (and dynamic conservation) is reflected by your findings? And, really, what is 'dynamic conservation'? I think you need to explain this statement better. Your subsequent sentences are meatier and help explain what you mean; you could simply delete this sentence and expand upon the following sentences to demonstrate some of these ideas to the reader.

• Response: Dynamic conservation is defined as similar to adaptive management within the cited paper. Nonetheless, we removed this citation and revised our recommendations for long term investments to have more clarity and relevance to our results. 

L619-623: Tangible recommendations...good stuff!

• Response: We revised the conservation implications to have more clear recommendations that are presented up front.

L623: As above, I'm not sure you really do show such a connection. You see a confirmatory shift in numbers in some species, not others....but you don't actually track birds--which is fine, that's a whole other study. But, you could cite the interconnections of these wetlands that others have demonstrated. If I'm not mistaken, others have conducted radio telemetry that demonstrates some of these connections (Takekawa, Warnock, Sanzenbacher, Evans Ogden, Shepherd, etc), some in SF Bay and Central Valley, others in similar systems (Fraser River).

Some useful references:

Sanzenbacher & Haig 2002 Waterbirds

Sanzenbacher 2001 (Pete's MSc.)

Warnock et al. 2004 CJZ (and lots of stuff from SF Bay with Sarah, showing movement patterns within bay...)

Pippa Shepherd and Leslie Evan Ogden's work at Simon Fraser...I think they showed habitat switching between estuaries and ag fields for WESA and DUNL?

Shepherd et al. 2003: Integrating marine and terrestrial habitats in shorebird conservation planning Wader Study Group Bulletin

etc.

I think some of these papers can be cited to a) demonstrate the vagility of shorebirds and b) actually link the various habitats of interest by individuals, info that would help you truly 'connect' your results...I agree with you that birds probably bailed on the Central Valley and went in search of new habitats (eg, Dunlin to the coast)...your surveys suggest this...and others have demonstrated these links (or at least such wanderings) with

• Response: The authors greatly appreciate these suggestions. Several of the references have been included in the revision to further elucidate past research that demonstrates how shorebirds respond to changing environmental conditions by moving among habitats and regions. In particular, we used this comment to supplement two paragraphs in the introduction. One of these paragraphs delves into movement between and among habitat types of the Central Valley and coastal regions. The reviewer may have overlooked this portion of the introduction, but regardless, the feedback was helpful to motivate us to better highlight these linkages.

---

## [Editor Report · Decision Letter 1]

6 Oct 2020

Effects of drought on the abundance and distributions of non-breeding shorebirds in central California, USA.

PONE-D-20-20377R1

Dear Dr. Barbaree,

We’re pleased to inform you that your manuscript has been judged scientifically suitable for publication and will be formally accepted for publication once it meets all outstanding technical requirements.

Kind regards,

Stephanie S. Romanach, Ph.D.

Academic Editor

PLOS ONE

Additional Editor Comments (optional):

Table 1 is clipped so not all values can be read in the current layout.
---

## [Editor Report · Acceptance letter]

12 Oct 2020

PONE-D-20-20377R1 

Effects of drought on the abundance and distribution of non-breeding shorebirds in central California, USA. 

Dear Dr. Barbaree:

I'm pleased to inform you that your manuscript has been deemed suitable for publication in PLOS ONE. Congratulations! Your manuscript is now with our production department. 

Kind regards, 

on behalf of

Dr. Stephanie S. Romanach 

Academic Editor

PLOS ONE